# Soundscapes and deep learning enable tracking biodiversity recovery in tropical forests

Jörg Müller [1,2] ✉, Oliver Mitesser[1], H. Martin Schaefer[3], Sebastian Seibold [4,5], Annika Busse[6], Peter Kriegel[1], Dominik Rabl [1], Rudy Gelis[7], Alejandro Arteaga[8], Juan Freile[9], Gabriel Augusto Leite[10], Tomaz Nascimento de Melo[10], Jack LeBien[10], Marconi Campos-Cerqueira [10], Nico Blüthgen [11], Constance J. Tremlett[11], Dennis Böttger [12], Heike Feldhaar[13], Nina Grella [13], Ana Falconí-López[1,14], David A. Donoso [14,15], Jerome Moriniere[16] & Zuzana Buřivalová [17]

Tropical forest recovery is fundamental to addressing the intertwined climate and biodiversity loss crises. While regenerating trees sequester carbon relatively quickly, the pace of biodiversity recovery remains contentious. Here, we use bioacoustics and metabarcoding to measure forest recovery post-agriculture in a global biodiversity hotspot in Ecuador. We show that the community composition, and not species richness, of vocalizing vertebrates identified by experts reflects the restoration gradient. Two automated measures – an acoustic index model and a bird community composition derived from an independently developed Convolutional Neural Network - correlated well with restoration (adj-R² = 0.62 and 0.69, respectively). Importantly, both measures reflected composition of non-vocalizing nocturnal insects identified via metabarcoding. We show that such automated monitoring tools, based on new technologies, can effectively monitor the success of forest recovery, using robust and reproducible data.

Tropical forests play a key role in the global carbon cycle and are central to Nature-based Climate Solutions, both in terms of climate adaptation and mitigation[1–4]. They are also fundamental to global biodiversity conservation, harbouring 62% of terrestrial vertebrate species[5]. As such, restoring tropical forests is key to counteract two of the major crises of our times, biodiversity loss and climate change. With ambitious goals such as the New York and Glasgow Leaders' Declarations on Forests[6], large-scale restoration projects are becoming increasingly common[7]. Yet, their success is far from guaranteed and often controversial[8]. Carbon storage and forest structure can be restored within several decades if appropriate species are planted and providing deforestation drivers have been addressed[9]. However, the recovery of tropical forest fauna varies widely, and depending on taxa, it is contentious and less predictable[9–13]. The complexity of

ecosystem processes, diversity of species, legacies of past land use, and the idiosyncrasies of conservation complicate precise and generalizable predictions of restoration success. Therefore, monitoring the performance of individual restoration projects remains key to adaptive management and evidence-informed conservation funding[14].

To be effective, all conservation measures require cost-efficient and robust biodiversity monitoring, which is lagging behind carbon monitoring due in part to the lack of scalable, reproducible and cost-effective sampling methodologies[13,15]. In particular, market-based conservation mechanisms that may rely on forest restoration, such as payments for ecosystem services, biodiversity offsets and credit markets, as well as e.g. forest sustainability certification[16], urgently require a cost-effective, transparent and generalizable biodiversity measurement and monitoring tool. Such a tool should facilitate

scalability in alignment with UN targets and help prevent green-washing: without the requirement and tool to monitor biodiversity, carbon-focused actors may plant simple, monoculture plantations, instead of forests that have the potential to become biodiverse and resilient with proper restoration.

Many taxonomic groups, including amphibians, birds, mammals, and insects include a considerable proportion of species that vocalize or otherwise use sound to communicate, making acoustic monitoring of these groups a particularly promising tool for biodiversity responses[17,18]. Sound diversity, expressed via the Soundscape Saturation Index, declined and became less synchronized with forest fragmentation and loss in Papua New Guinea and Borneo[19,20]. In Puerto Rico, soundscapes became impoverished and subsequently recovered after a major hurricane, reflecting the reassembly of the vocalizing fauna[21]. Likewise, bird vocalizations and soundscapes were altered with forest degradation from selective logging in Southeast Asia[22]. These studies show that forest loss and degradation can be tracked by using soundscapes, but it remains unclear if soundscapes reliably track the restoration of faunal biodiversity in tropical forests[23].

Many acoustic indices have been developed to reduce the complexity of multidimensional, information-rich soundscapes to an interpretable level[24,25]. Such indices describe different aspects of the soundscape, from signal-to-noise ratio, variation in frequency, to complexity and entropy[25,26]. Currently, no single best index exists, as most indices have been tested only in a handful of habitats or land use contexts, and when used individually, they often yielded mixed results[27,28]. For example, some studies observed a strong correlation between the Acoustic Complexity Index, a measure for biotic activity[29], and both the number of avian vocalizations and species richness[24,29,30], while others reported inconsistent results[31,32] or no correlation[33,34]. We hypothesize that a combination of indices may best explain the composition and richness of the recovering vocalizing animal community, as some indices focus on the sound coverage along the frequency range (Soundscape Saturation) whereas others quantify the sound diversity over time (Acoustic Complexity Index, Temporal Entropy, Entropy of Frequency) or simply the activity (Events/Second).

Apart from acoustic indices, techniques for discerning specific animal species from soundscape recordings are also being developed[35]. Out-of-the-box models for species identification are typically less data hungry, computationally simpler in design (e.g. relying on MCMC vocal separators), but depend on human-guided feature engineering (i.e. "supervised machine learning"), introducing potential subjectivity that could hinder performance, particularly with diverse or noisy datasets. More recently, artificial intelligence deep learning models, such as Convolutional Neural Networks (CNN), have been developed to identify birds, bats or amphibians[36–40]. They tend to be more flexible and may require fewer person hours to generate well-performing models. When such models become available for many species in a region, species communities could be determined automatically, which has however not been sufficiently investigated to date. A major bottleneck for machine learning approaches, including CNNs, is the need of large training datasets. This is particularly challenging in the hyper-diverse communities of tropical forests[41]. Yet, it is increasingly feasible to identify entire vocalizing communities to species level due to the accumulation of labelled acoustic datasets and progress in the development of tools (e.g. Arbimon, BirdNET[40,42]). Hence, the combination of innovative methods may allow encompassing biodiversity assessments across multiple sites, leveraging community, functional, and phylogenetic diversity measures. As community composition has been shown to predict environmental gradients[43,44], we hypothesize that community composition derived from CNN models will be correlated with recovery gradients in tropical forests.

To pioneer the assessments of biodiversity recovery in tropical forests, we sampled soundscapes simultaneously with the same devices and protocol in 43 plots along a recovery gradient in a space-for-time substitute approach (Fig. 1). Our plots in the Ecuadorian lowland Choco comprised active cacao plantations and pastures, abandoned cacao and pastures with forest recovery for 1–34 years, and old-growth forests. We carried out (i) manual, expert-led identification of vocalizing birds (183 species), mammals (3 species, excluding bats) and amphibians (41 species) within simultaneous time windows covering 28 min in all plots across two days, (ii) an acoustic index analysis for two weeks of recordings including the days analyzed by the experts, and (iii) for the same time span a presence and absence of 75 selected bird species identified with a CNN model, trained on an independent dataset. To test the generality of our results, i.e., that community composition indeed tracks faunal recovery, we (iv) used a sound-independent dataset by sampling nocturnal insects with autonomous light traps and assessed insect diversity using meta-barcoding. As fewer than 1% of species were non-Insecta, herafter we refer to this dataset as insects. Insect diversity, in general, has been shown to correlate with bird, frog and mammal diversity[45]. Our results demonstrate that automated bioacoustics monitoring can be used to track tropical forest recovery of animal communities from agricultural abandonment beyond vocalizing vertebrates, suggesting its broad use to assess restoration outcomes.

## Results and discussion

We first investigated how well the total vocalizing vertebrate species community identified by experts represents the forest recovery gradient. We then modelled the community composition, species richness, the richness of species observed in our old-growth plots (old-growth species hereafter), and the composition of nocturnal insects, using the two data types derived from soundscapes: a set of acoustic indices and the first principal component (PC) of a CNN-derived bird community NMDS map (first community axis, hereafter). As the main axis of community composition (Fig. 2a) revealed a linear gradient from pastures to old-growth forests, we used linear models in all further analyses as the most parsimonious approach.

### Vocalizing vertebrate communities represent the recovery gradient

At our study sites in the Choco, Ecuador (Fig. 1), the community of vocalizing vertebrates, as identified by experts, showed a clear gradient along the first axis of a non-metric multidimensional scaling (NMDS) ordination (Fig. 2a). Abandoned cacao plantations initially recovered vertebrate diversity faster than abandoned pastures, but converged during later succession (Fig. 2a). Most old-growth forests were rather distinguishable in community composition from regenerating sites along the first ordination axis, with the exception of one pasture regenerating for 34 years (Fig. 2a). This is in contrast to the earlier broader overlap of bird communities in regenerating tropical dry forests with old-growth reference sites[46]. Total species richness steadily decreased along the recovery gradient in line with one study in tropical dry forests[47], but in contrast to another study on birds[46]. One possible explanation for this high richness in agriculture plots might be that more species transit through these open sites. Another explanation might be spill-over from surrounding forests in the small-scale agriculture of our study region, supported by old-growth species also in pastures (Fig. S1). In contrast, the richness of old-growth-specific species increased over time (Fig. S1).

The positive correlation between recovery time and community composition may raise the question of why not just use time elapsed as a proxy for biodiversity recovery, such as sometimes done with carbon accumulation[13]. From our results, we conclude against simply using time for two reasons. First, we found a substantial overlap of species composition between the early and late recovery stages, with a most pronounced change in the early phase. This demonstrates a rapid shift of species compositions after abandonment and a slowing down of

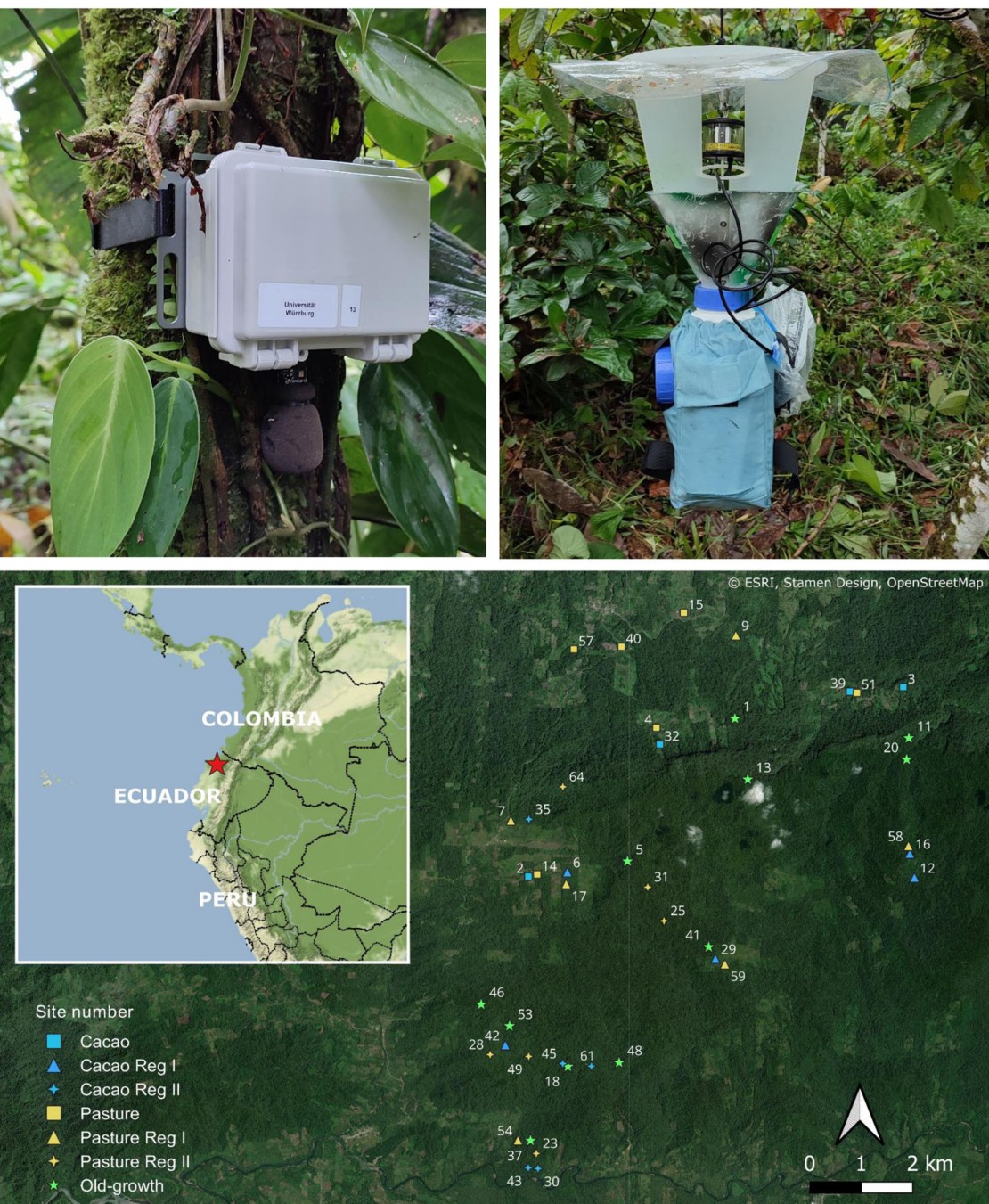

**Fig. 1 | Map of the study area, the sampling locations and sampling devices to test soundscapes and metabarcoding for monitoring of restoration success in tropical forests; top left sound recorder, top right light trap.** The map was created in QGIS using the ESRI "Satellite" basemap (Scale Not Given, January 23rd 2023, https://server.arcgisonline.com/ArcGIS/rest/services/World_Imagery/Map-Server/tile/{z}/{y}/{x}) and the "Stamen Terrain Background" and "Stamen Terrain Lines" basemaps (Scale Not Given, January 23rd 2023, https://maps.stamen.com/.

community changes in the later stages. Furthermore, the high variation in the early stages of recovery in our study systems might be affected by variation in forest cover in the surrounding as recently shown as relevant for predictability of forest regeneration[48]. Second, additional anthropogenic impacts as logging or hunting[20,49] can strongly affect the local fauna, but are not represented by recovery age.

### Sound diversity as measure of biodiversity
To assess which type of information derived from soundscapes best reflects the faunal recovery at our sites, we estimated the explanatory

power of (i) a combination of five selected acoustic indices and (ii) the first community axis of a CNN-derived bird community in four multiple regression models with the following diversity response variables, determined by expert analysis: (1) the first community axis of species composition of vocalizing vertebrates, (2) total species richness, (3) richness of old-growth species, and (4) the first community axis of nocturnal insect communities (Table 1). Acoustic indices individually had low explanatory power, but we found that a combination of acoustic indices had a high explanatory power for the vertebrate community composition (adj. $R^2$ = 0.62), good explanatory power for richness of old-growth species and nocturnal insect composition, but

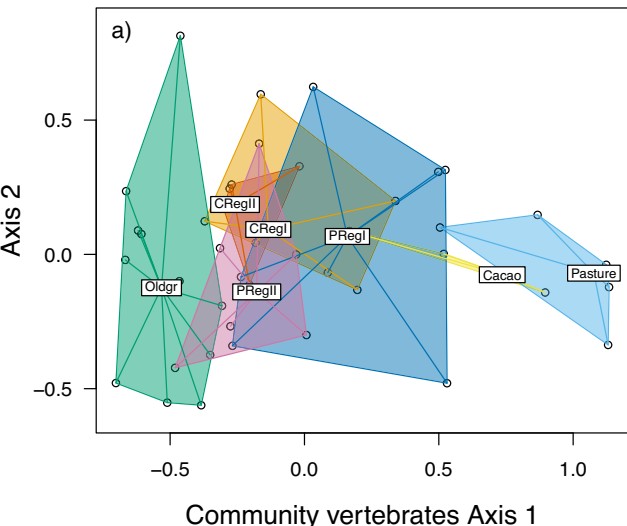

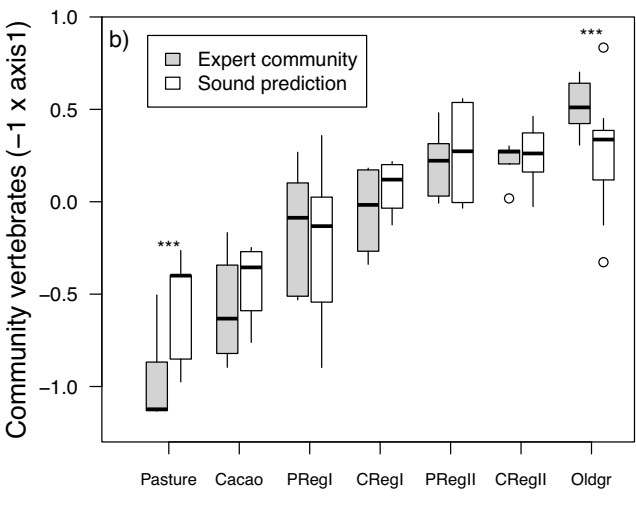

**Fig. 2 | Community composition of vocalizing vertebrates identified by experts from audio files and predictions from acoustic indices. a** NMDS plot shows a shift in the species composition along the recovery gradient (right to left, colours display land-use and recovery stage as well as old-growth category); **b** First axis of observed (expert identification) and predicted (four acoustic indices, Table 1) values of community composition. Pasture (P) and Cacao (C) = active agriculture plots, Reg I = early regeneration forests (1–19 years), Reg II = late regeneration forests (20-37 years), and Oldgr = old-growth forests. Asterisks(***) indicate highly significant difference ($p < 0.001$) between both methods within a category of the recovery gradient based on a linear mixed-effects model ($n = 43$ biologically independent plots). The boxplots show the median, and the 25th and 75th percentile, the whisker show the minimum and maximum values not exceeding a distance of $1.5 \times$ interquartile range, values beyond are plotted as single points.

low explanatory power for total vertebrate richness (Table 1). Our findings add to the growing literature that shows that using a set of acoustic indices instead of single ones is better for estimating biodiversity[50].

Despite the potential of acoustic indices as predictors of vocalizing vertebrate diversity, these do not distinguish between late recovery and old-growth communities, as seen by comparing observed versus predicted values in Fig. 2b. This lack of sensitivity could be partially due to acoustic indices saturating at high diversity values[16,51]. For example, species in functionally richer landscapes may have smaller acoustic 'niches', due to competition for acoustic space[26]. This could result in additional species not substantially changing index values.

In the next step, we applied an independent artificial intelligence model for bird species identification, developed and trained in the region of our study prior to our sampling. Despite the model identifying only ~25% of species detected by the experts in our data, the model-derived first community axis was the single best predictor for expert-derived community composition. With adjusted $R^2$ of 0.69, it outperformed the combination of acoustic indices, demonstrating that community composition estimated from CNN is a promising way to track faunal recovery of tropical forest communities (Fig. 3a). Including a higher proportion of vocalizing species, including amphibians and mammals, would presumably further increase our ability to track faunal recovery through sound. This conjecture is supported by an even higher explanatory value ($R^2 = 0.85$) of community composition when restricting birds species to only those represented in both the expert-led and the CNN identification ($n = 49$, Fig. 3c).

Crucially, both automated sound-based measures (indices and CNN) had a good explanatory power even for species that are not part of the vocalizing tropical animal community, such as the largely silent nocturnal insects (Fig. 3b). In further testing, we excluded the 13 vocalizing insect species from the families *Tettigoniidae, Gryllidae, Cicadidae*, present in our light traps, which did not change the axis of species composition (Pearson correlation rho=0.986), indicating that light trap assemblages predominantly represent non-vocalizing insects. Acoustic indices might thus serve as powerful surrogates for

tropical faunal biodiversity including important functional groups, such as pollinators or decomposers. Our results demonstrate that both products from soundscapes recordings and analyses—a set of acoustic indices and CNN-derived species composition—are suitable for tracking faunal recovery across a few decades. Moreover, we show that measures based on composition are more powerful than richness alone, as known for insect communities in tropical forests[52].

As vocalizing vertebrate communities progressed towards old-growth forest conditions, the soundscapes showed increasing Soundscape Saturation and Entropy of Frequency, but decreasing Acoustic Complexity and Events per Second from pastures to old-growths (Fig. 4 and Table 1). The latter concurs with the observation of increasing acoustic complexity with increasing number of species[24,29,30]. In our dataset, we found more species and more vocalizations in agricultural plots (Fig. 4 and Fig. S2). More species might transit through these open sites, or this might be due in part to a higher vertical structure complexity of old-growth forests: our recorders were always placed about 2 metre above ground, possibly underestimating canopy species particularly in old-growth plots with very large trees. Additionally, denser vegetation in old-growth forests may limit sound propagation[53]. However, bird call playback experiments in other countries revealed similar patterns for different recovery stages in tropical forests, which supports similar probabilities of recording[47].

However, we would also expect attenuation biases in recovering forests, which have higher stem density[54]. Overall, we therefore think that the sound attenuation bias is minimal in our study system. The increase of Soundscape Saturation along the recovery gradient (Fig. 4) could indicate that more mature forests have an acoustically more diverse communities, resulting in a broader coverage of the frequency spectrum. This is in line with the observed decrease of Soundscape Saturation with fragmentation[19].

The total richness of vocalizing vertebrates decreased with Soundscape Saturation and increased with Events/Second, both in line with fewer species recorded in old-growth plots (Table 1). The richness of species recorded in old-growth plots increased with the Entropy of Frequency and decreased with Acoustic Complexity reflecting low species richness but a functional highly diverse community in old-

**Table 1 | Acoustic indices as predictors of scores of the first axis of ordination of vertebrate communities, the richness of vertebrates per plot, only vertebrates observed in old-growth plots, and the scores of the first axis of an ordination from light trapping insect communities using a multiple linear regression model (see "Methods" for explanations of acoustic index calculation)**

| | Response variables | | | | | | | |
|---|---|---|---|---|---|---|---|---|
| | **Vertebrate community axis 1** | | **Vertebrate richness** | | **Vertebrate old-growth richness** | | **Nocturnal insect communities axis 1** | |
| **Adjusted $R^2$** | **0.62** | | **0.20** | | **0.45** | | **0.42** | |
| **Acoustic indices (predictor variables)** | **Est. ± SE** | **t-val.** | **Est. ± SE** | **t-val.** | **Est. ± SE** | **t-val.** | **Est. ± SE** | **t-val.** |
| Temporal entropy | 3.7 ± 2.2 | 1.66 | −2.4 ± 1.8 | −1.37 | 0.3 ± 1.7 | 0.19 | 0.0 ± 1.2 | 0.01 |
| Acoustic complexity | −23.7 ± 8.0 | **−2.95** | 9.8 ± 6.5 | 1.51 | −14.3 ± 6.1 | **−2.36** | −5.2 ± 4.4 | −1.18 |
| Entropy of frequency | 3.7 ± 1.3 | **2.94** | −0.8 ± 1.0 | −0.74 | 2.0 ± 1.0 | **2.09** | 1.5 ± 0.7 | **2.11** |
| Soundscape saturation | 3.2 ± 0.7 | **4.54** | −1.6 ± 0.6 | **−2.90** | 0.9 ± 0.5 | 1.78 | 1.2 ± 0.4 | **3.06** |
| Events/Second | −0.8 ± 0.2 | **−3.62** | 0.3 ± 0.2 | **2.04** | -0.3 ± 0.2 | −1.69 | −0.3 ± 0.1 | **2.33** |

Richness values were log-transformed and modelled with the same linear models to make adj. $R^2$ comparable. Estimates, standard errors and $t$-value are shown; bold $t$-values indicate significant predictors ($p < 0.05$). For spatial independence of all model residuals see Supplementary material.

growth forests (Table 1). With the community composition of insects shifting towards old-growth forests, Entropy of Frequency, Soundscape Saturation and Events per Second all increased.

The communities of vocalizing vertebrates, as well as the community of nocturnal insects, were highly correlated with the main axis of species composition derived from CNN (Fig. 3). Our data show not only that nocturnal insects recover quickly with forest regeneration but that CNN models track that recovery well. Species composition, rather than species richness, thus serves as an indicator of forest recovery. This is in line with a recent meta-analysis using a global data set of audio-recordings paired with manual avifaunal point counts[28]. Even here, acoustic indices were not explained simply by species richness, but soundscape changes indicated changes in community composition. We therefore posit that more emphasis should be placed on species composition rather than solely on species richness, and that species composition identified from sound via experts, or now increasingly via artificial intelligence, is a powerful tool that can be substituted by sound indices where species-based analyses are not available. Specifically, the vocalizing vertebrate communities, sampled by audio recorders, can therefore serve as indicators for the faunal recovery of tropical forests.

In summary, our results show that soundscape analysis is a powerful tool to monitor the recovery of faunal communities in hyper-diverse tropical forest. Soundscape diversity can be quantified in a cost-effective and robust way across the full gradient from active agriculture, to recovering and old-growth forests. The promising results from our artificial intelligence application, albeit with only 25% of the bird species that were identified by experts, show the potential of automated identification of species communities from sound data. Importantly, we document that acoustic indices track the quick and consistent biodiversity recovery even in complex, hyper-diverse ecosystems such as tropical forests, independent of land use legacies. To generalize the soundscape approaches, Artificial intelligence-models for vocalizing animals have to be improved globally and their application to environmental gradients as the recovery gradient in our study has to be validated regionally. Therefore, we urge the conservation community to prioritize the creation of global sound repositories for taxa beyond birds, based on which machine learning models can be rapidly improved and extended. Implementing passive, soundscape-based biodiversity monitoring in tropical ecosystems without relying on expert knowledge would allow conservation managers to assess forest recovery cost-effectively and to better quantify the conservation value of their protected areas. The standardized collection of raw sound environmental data, such as soundscapes, creates a reproducible comprehensive long-term data basis in biodiversity monitoring that is easier to store in the long term than many

specimen collections and largely independent of the collector. This, in combination with making soundscape data publicly accessible, could also help to reduce greenwashing in carbon-focused conservation. Being able to directly quantify biodiversity, rather than relying on proxies such as growing trees, encourages and allows external assessment of conservation actions, and promotes transparency. Furthermore, well-documented datasets can be re-analyzed retrospectively using the latest biostatistical methods. They can empower practitioners and funders to quantify the biodiversity gains in regenerating tropical forests, a crucial prerequisite for monetizing biodiversity objectives associated with carbon removal, allowing markets to address the joint biodiversity and climate crises of our times. Restoring tropical forests so that they provide carbon and biodiversity benefits will increase the likelihood that nature-based solutions will result in resilient, biodiverse ecosystems and not empty carbon farms.

## Methods
### Study site and design
We selected 43 study plots (Fig. 1) within the research unit REAS-SEMBLY (www.reassembly.de)—a collaborative Ecuadorian-German research approach to study predominantly the shift of species networks along a tropical recovery gradient. Our plots followed a chronosequence of forest recovery, and included (i) active pastures and cacao plantations; (ii) formerly used pastures and plantations, which were secondary forests regrowing for 1 to 34 years at the time of the study, and (iii) old-growth forests, with no indication of recent use by humans. The plots were between 159 and 615 m a.s.l. All recovery processes were natural after abandonment, without active planting.

### Audio data collection
In October 2021, we deployed one Bioacoustic Recorder (BAR-LT, Frontier Labs, Meanjin, Australia) with one omnidirectional microphone facing down at a height of ~1.70 m above ground (Fig. 1), at each of the 43 plots. The recorders were programmed to record 2 min every 15 min throughout the day for two weeks (Julian day 299-314 in 2021) concurrently, with a sampling rate of 44.1 kHz.

### Expert-based community composition
Experts identified birds and mammals from 2-minute files recorded at 06:00, 06:30, 07:00, 12:00, 16:00, 17:00, 18:00 h from 2 days without heavy rain, covering the high activity phases of birds and vocalizing mammals (howler monkeys) around dusk and dawn, as well as few minutes during the day to cover flock activities. Additional bird data were evaluated at 06:15, 06:45, 07:15, 12:15, 16:15, 17:15, 18:15 h. Amphibian data were evaluated in 2-min time windows starting at

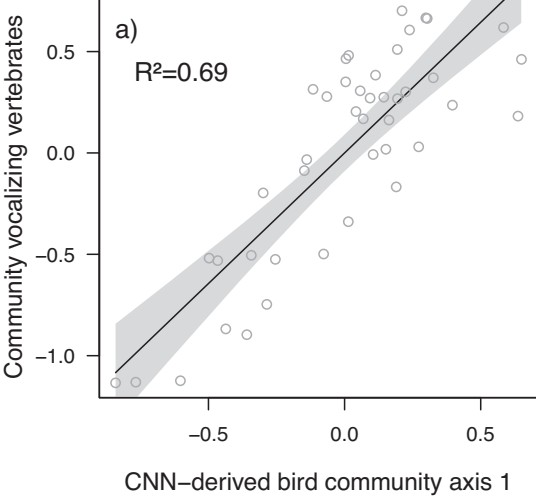
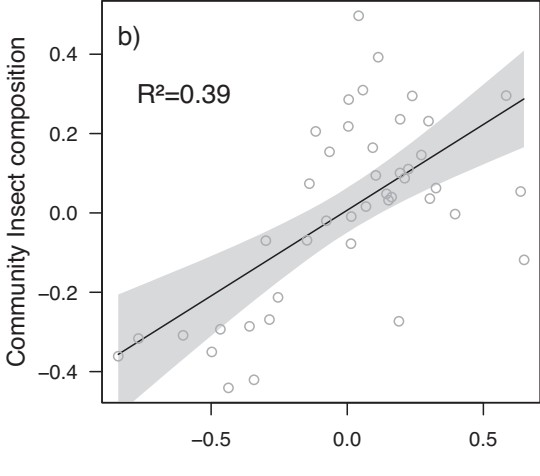
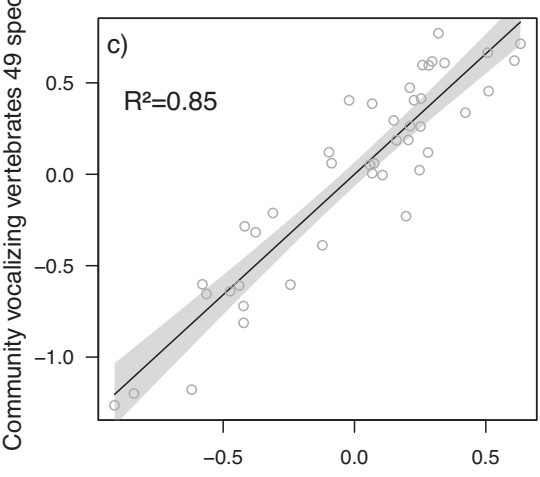

**Fig. 3 | CNN-derived bird community composition as predictors for birds and insect communities.** CNN-bird community versus (**a**) first community axis of total vocalizing vertebrates by experts (the scores of the first NMDs axis in Fig. 2a and all other ordinations were mirrored to create an ascending recovery gradient from pastures to old-growth forests), versus (**b**) first nocturnal insect community axis by metabarcoding of light trap samples along the forest recovery gradient and versus (**c**) Expert-derived bird community composition based on 49 species shared in expert and CNN predictions. $R^2$ values represent adjusted-$R^2$ from a linear regression; polygons represent 95% confidence intervals of the means of the linear regression.

00:00, 03:00, 05:00, 09:00, 18:45, 20:30 h reflecting times of their peak activity after expert evaluation of some selected plots over the whole day. Here we used one day with rain and one without rain (Fig. S1).

The metric "frequency per plot" was calculated by counting files containing a recording of a species; for more details see Supplement. Mammal and bird species were identified by one of us (R.G.). To assess inter-observer bias, additional sound files shifted by 15 min were evaluated by another bird expert (J.F.). Both datasets provided similar results (Fig. S3).

**Acoustic index calculation**
Acoustic indices were calculated by using the toolbox "AnalysisProgram.exe"[55] for each 2-min recording separately on the high performance cluster of the University of Würzburg. Soundscape Saturation, Acoustic Diversity Index, Bioacoustic Index, and Acoustic Eveness were calculated from Towsey's amplitude spectrum according to Burivalova[20] and Ross[56]. We used the R-package "stringr" for data pre-processing[57] in R version 3.6.3. From a number of established

soundscape characteristics[24], we selected five independent indices representing different aspects of sound diversity (Table 1).

**CNN-based community composition**
For automated species identification, we applied a multi-label audio recognition model. The algorithm we used was developed independently from this study in the framework of Arbimon[42]. The model had been previously trained to recognize 115 common song classes produced by 112 species, of which 77 might occur in our study region (Supplementary Data 1). Data used to train the CNN model came from 401,685 1-min soundscape recordings previously collected in a separate study from 55 sites in and around Mashpi Rainforest Biodiversity Reserve and Canandé Reserve in the Ecuadorian Chocó Forest (500–2300 m a.s.l.). Recordings for training were collected between January and December 2019, using Audimoth devices[58] and Guardian recorders (https://rfcx.org/guardian). Audiomoths were placed on a tree at the height of 1.5 m and programmed to record 1 min of audio every 10 min for a total of 144 recordings per day at a sampling rate of 48 kHz. Guardian recorders were deployed at the canopy of trees at a

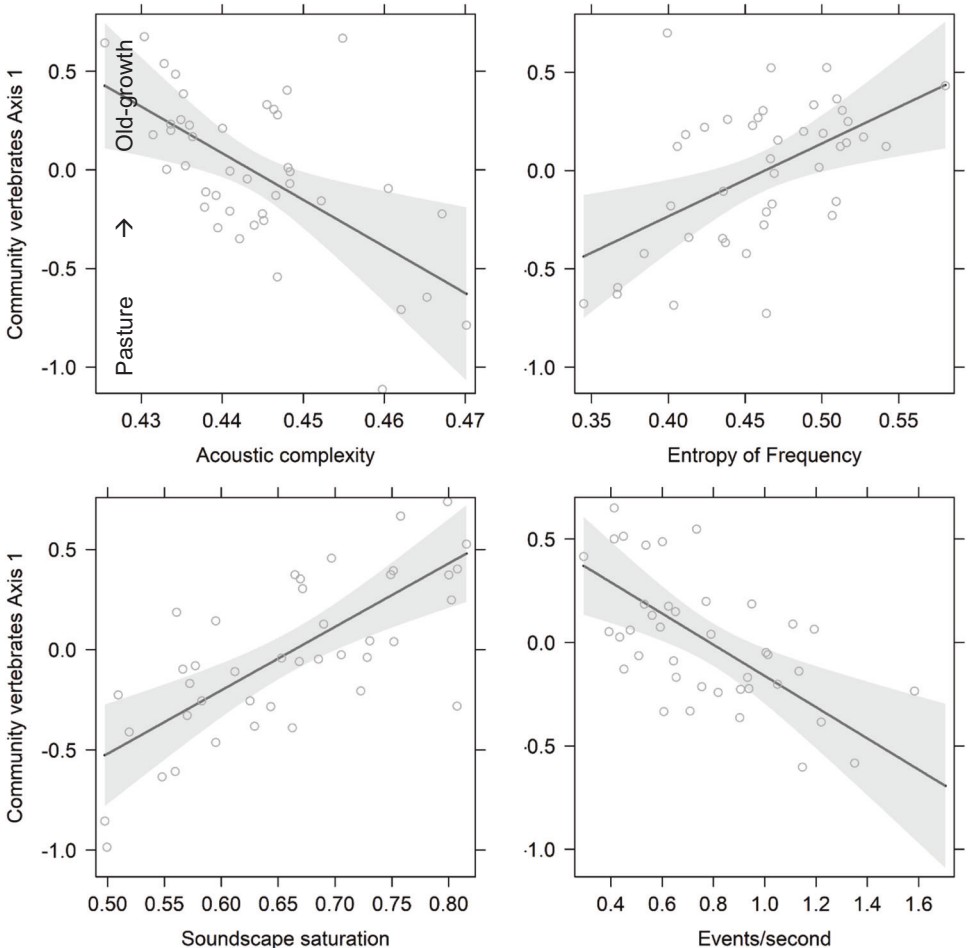

**Fig. 4 | Partial effect plots for four acoustic indices explaining the community composition of vocalizing vertebrates in 42 plots along the recovery gradient from pastures to old-growths (the inverse axis of the NMDs axis in Fig. 2a), for** statistics of the full model see Table 1. Top left the partial effect of acoustic complexity, top right of entropy of frequency, down left of soundscape saturation and down right of event per second.

height of ~30 m and programmed to record continuously at a sampling rate of 12 kHz.

Training data for the audio recognition model was created using the method described in LeBien et al.[59]. The initial training dataset consisted of a set of positive and negative samples for each song class, collected using a template matching analysis. The positive and negative samples can be considered true positives and false positives from a rudimentary single-class detector, respectively. Thus, for each training sample, the presence or absence of only a single class is known. However, it was assumed that each class is only present in its positive samples. This was expected to greatly increase the amount of information in the training data by introducing a large amount of true negative labels, at the expense of possibly introducing a comparatively insignificant amount of false negative labels. The audio training data was upsampled to 16 kHz if needed. Spectrograms were computed from three-second audio segments using a short-time Fourier transform (STFT) with an STFT window length of 0.1 s, a hop length of 0.025 s, a frequency range of 50Hz to 8 kHz and 224 frequency bins. Spectrogram amplitudes were log-scaled, and mel-scaling was applied to the frequency axis.

Data augmentation was applied by combining random pairs of samples in each training data batch. This increases variation in the training data, and allows control over the number of samples per class for multi-label training. For each sample $A$ in a batch, another sample $B$ is randomly selected. The spectrogram of sample $B$ is cropped in frequency to the band containing the labelled species' call (i.e. the single

species with known presence or absence). The frequency band of each class was estimated visually by experts during data labelling. Sample $B$'s spectrogram is then blended with the same frequency band of $A$'s using an element-wise maximum operation. The labels of $A$ and $B$ are combined with the same operation. All song classes with at least 10 positive training examples were included in the model. The number of positive samples collected per class was highly imbalanced, ranging from <10 to >1000. Random resampling was applied to compensate for the data imbalance. The target number of positives per class was set to an intermediate value of 200. The target number of negatives was set at a moderate value of 50 to avoid a very low ratio of positive to negative training labels per class, considering the assumption applied to the labels.

The final layers of the MobileNetV2 model that learn a predictive model from the extracted convolutional layer features were replaced. Our model's final layers consisted of an average pooling of the extracted features, a dropout layer, and a dense layer with a sigmoid activation to produce the output scores. Binary cross-entropy loss and the Adam optimizer[60] with an initial learning rate of 0.001 were used. The model was trained for one hundred epochs with a batch size of 16. The model was validated against a set of 200 one-minute audio files, each labelled with the set of present classes. The best model weights were chosen based on the best class-wise mean average precision on the validation dataset. The model was finally evaluated against a dataset including the validation set and an additional, independent 200 expert-labelled one-minute recordings from similar habitats as in

our study (Supplementary Data 1). The mean average-precision across the classes of the best model was 86.9%. This score represents the average precision across all thresholds, using the changes in recall between each consecutive threshold as weights for the average. At an output score threshold of 0.5, the model achieved a mean F1-score across classes of 77.2%, a mean precision of 82.9% and a mean recall of 80.7%. For a threshold of 0.8, the mean F1-score, precision, and recall were 78.9%, 89.4%, and 77.5%, respectively. Supplementary Data 1 contains species-specific scores for each of these metrics. There is no strong bias toward the most common classes (Pearson correlation $\rho = 0.07$, $p = 0.48$, Precision 0.8 - log(Presences)).

The model takes a three-second waveform as input and outputs an independent score for each song class. The output score represents the model's confidence that the class is present in the sample. First, input sound clips are converted to spectrogram images for feature extraction. A MobileNetV2 convolutional neural network (CNN) pre-trained on the ImageNet dataset was fine-tuned on spectrograms of an independent audio training data[61]. Models pre-trained on ImageNet are configured to extract a variety of generic image features that can be useful for spectrogram recognition[40]. In comparison with randomly initialized models, models fine-tuned with transfer learning generally require less time and training data for strong performance. While pre-trained models for audio recognition have recently been proposed (YAMNet, VGGish[62]), they are trained on sounds of limited time duration and resolution (rather than on images of sounds), which can impact their applicability to noisy and complex signals such as found in biodiverse soundscapes. Random resampling and data augmentation were applied to compensate for data imbalance and improve model generalization. The algorithm evaluated each input audio file in our study by scoring every three-second window of audio with a one-second shift between window start times. With this model, we identified 73 species in our 43 plots (Tab. S1). Next, we predicted species presence in each file based on a 0.5 and 0.8 output score threshold for each of the 73 species. Both revealed very similar species communities, therefore, we used the more conservative threshold of 0.8 to declare a species as present in a specific file.

## Light traps

During the same period of the sound sampling, we set up an autonomous light trap for one night per plot. These traps were equipped with an LED light optimized for insect sampling powered by a Power bank (LepiLED Mini Switch 0.65, UV-mode switched off, Brehm, Jena Germany[63]). This kind of light attracts predominantly Lepidoptera and Dipteran. However, with the wide range of species attracted, it is one of the most efficient methods for tropical nocturnal insects, even collecting some vocalizing insect species as cicadas. For 8 h after dusk, insects were collected in a jar mounted under a funnel and killed by chloroform. Insect collections were done following Ecuadorian Laws under Contrato Marco MAE-DNB-CM-2019-0115 and Export Permit 007-2022-EXP-CM-FAU-DBI/MAAE.

We removed large-bodied Lepidoptera from the Saturnidae and Sphingidae families (moths) as well as Coleoptera (beetles) individuals for taxonomic identification. The remaining insect bulk was frozen and transferred to 96% undenatured ethanol. The samples were then passed through an 8-mm sieve, thereby separating larger and smaller insects, opening the avenue for using read numbers for abundance estimates[64]. This was done to increase the likelihood of detecting small-bodied and rare species in samples, as individuals with a larger biomass provide disproportionate amounts of DNA and can therefore be over-represented when metabarcoding bulk samples[65]. Size filtering is only one tool to improve the balance between small and rare species on the one hand and large and abundant species on the other hand in bulk samples, and even this cannot guarantee all insects be detected. Here even species belonging to the same genus may be differently impacted by the same DNA

extraction/amplification/sequencing procedure, generating a difference in read numbers, which may result in loss of rare species[66]. However, our approach was standardized for all samples. The CO1-5P (mitochondrial cytochrome oxidase 1) target region was sequenced for collected bulk samples, following the laboratory and bioinformatic pipelines reported in Hausmann et al.[67] and as described in the following.

## High-throughput sequencing

Preservative ethanol was removed and the mixed arthropod samples were dried overnight in a 60–70 °C oven to evaporate the residual ethanol. The dried arthropods were then homogenized with stainless steel beads within a FastPrep 96 system (MP Biomedicals). DNA was extracted from all samples by incubating them in a 90:10 solution of animal lysis buffer (buffer ATL, Qiagen DNEasy tissue kit, Qiagen, Hilden, Germany) and proteinase K. After an overnight incubation in a 56 °C oven, the samples were left to cool to room temperature. DNA was extracted from 200-µL aliquots using the DNEasy blood & tissue kit (Qiagen) following the manufacturer's instructions. Multiplex PCR was performed using 5 µL of extracted genomic DNA, Plant MyTAQ (Bioline, Luckenwalde, Germany) and high-throughput sequencing (HTS)-adapted mini-barcode primers targeting the mitochondrial CO1-5P region (mlCOIintF – 5′-GWACWGGWTGAACWGTWTAYCCYCC-3′; dgHCO2198–5′-TAAACTTCAGGGTGACCAAARAAYCA-3′; following Leray et al.[68]–also see Morinière et al.[69,70].

Amplification success and fragment length were determined using gel electrophoresis. The amplified DNA was cleaned and each sample was resuspended in 50 µL of molecular water. Illumina Nextera XT (Illumina Inc., San Diego, USA) indices were ligated to the samples in a second PCR, conducted at the same annealing temperature as in the first but with only seven cycles. Ligation success was confirmed by gel electrophoresis. DNA concentrations were measured using a Qubit fluorometer (Life Technologies, Carlsbad, USA), and the samples then combined into 40-µL pools containing equimolar concentrations of 100 ng each. The pooled DNA was purified using MagSi-NGSprep Plus beads (Steinbrenner Laborsysteme GmbH, Wiesenbach, Germany). The final elution volume was 20 µL. HTS was performed on an Illumina MiSeq using v3 chemistry (2*300 bp, 600 cycles, maximum of 25 mio paired-end reads).

## Bioinformatics

Paired-ends were merged using the -fastq_mergepairs utility of the USEARCH suite v11.0.667_i86linux32[71] with the following parameters: -fastq_maxdiffs 99, -fastq_pctid 75, -fastq_trunctail 0. Adaptor sequences were removed using CUTADAPT[71] (single-end mode, with default parameters). All sequences that did not contain the appropriate adaptor sequences were filtered out in this step using the --discard-untrimmed parameter. The remaining pre-processing steps (quality filtering, dereplication, chimera filtering, and pre-clustering) were carried out using the VSEARCH suite v2.9.1[72]. Quality filtering was performed using the --fastq_filter VSEARCH utility (parameters: --fastq_maxee 1, --minlen 300). Sequences were dereplicated with --derep_fulllength (parameters: --sizeout, --relabel Uniq), first at the sample level, and then at the combined dataset level after concatenating all sample files into one large FASTA file, which was also filtered for singletons (sequences occurring only once in the entire dataset and a priori considered as noise; parameters: --minuniquesize 2, --sizein, --sizeout, --fasta_width 0). To save processing power, a pre-clustering step (at 98% identity) was employed before chimera filtering using the --cluster_size VSEARCH utility with the centroids algorithm (parameters: --id 0.98, --strand plus, --sizein, --sizeout, --fasta_width 0, --centroids). Chimeric sequences were then detected and filtered out from the resulting file using the VSEARCH --uchime_denovo utility (parameters: --sizein, --sizeout, --fasta_width 0, --nonchimeras).

A custom perl script obtained from the authors of VSEARCH (see https://github.com/torognes/vsearch/wiki/VSEARCH-pipeline) was then used to regenerate the concatenated FASTA file, but without the previously detected chimeric sequences. The resulting chimera-filtered file was then used to cluster the reads into OTUs using SWARM v.3.1.0[73] (parameters: -d 13 -z). The value for the d parameter was chosen based on the experiments of Antich et al.[74]. The OTU representative sequences were then sorted using VSEARCH (parameters: --fasta_width 0 --sortbysize) and an OTU table was constructed from the resulting FASTA file using the VSEARCH utility --usearch_global (parameters: --strand plus --sizein --sizeout --fasta_width 0). To reduce the risk of false positives, a cleaning step was employed that excluded read counts in the OTU table constituting <0.01% of the total number of reads in the sample. OTUs were additionally removed from the results based on negative control samples, i.e. if the number of reads for the OTU in any sample was less than the maximum among negative controls, those reads were excluded from further analysis. OTU representative sequences were blasted (parameters: programme: Megablast; maximum hits: 1; scoring (match mismatch): 1-2; gap cost (open extend): linear; max E-value: 10; word size: 28; max target seqs 100) against (1) a custom database downloaded from GenBank (a local copy of the NCBI nucleotide database downloaded from ftp://ftp.ncbi.nlm.nih.gov/blast/db/), and (2) a custom database built from data downloaded from BOLD (www.boldsystems.org[75,76]) including taxonomy and BIN information, by means of Geneious (v.10.2.5 – Biomatters, Auckland, New Zealand). All available Animalia data was downloaded from the BOLD database on 29 July 2022 using the available public data API (http://www.boldsystems.org/index.php/resources/api) in a combined TSV file format. The combined TSV file was then filtered to keep only the records that: (1) had a sequence (field 72, "nucleotides"); (2) had a sequence that did not hold exclusively one or more "-" (hyphens); had a sequence that did not contain non-IUPAC characters; (3) belonged to COI (the pattern "COI-5P" in either field 70 ("markercode") or field 80 ("marker_codes")); (5) had an available BIN (field 8, "bin_uri"). In (5), an exception was made in cases where the species belonging to that record did not occur with a BIN elsewhere in the dataset. In other words, "BIN-less" records were kept if their species were also completely BIN-less in the dataset.

The overall aim was to obtain a matrix of taxonomic units that closely resembles the concept of species. Therefore, the COI sequences were used to attribute Barcode Index Numbers (BINs), which are clusters of barcode sequences that can be used as a proxy taxonomic unit. BINs avoid the situation that in certain lineages there are unequally more OTUs even within a species. The allocation to BIN units is a challenge, given today's still very incomplete libraries. In many regions, corresponding libraries are largely missing for large species groups. This is true especially for groups harbouring "dark taxa" such as dipterans, hymenopterans and hemipterans, but also for large portions of other arthropod species, which are currently not referenced for South America. For ecological analyses, the goal is to assign the sequences to units representing the solution of species themselves and to derive ecological properties from the sequence information. For this purpose, we have developed the following procedure. The sequences are assigned to the next existing BIN from the studied and neighbouring countries reporting the genetic distance. This proximity and the information to which family, genus or species the sequence belongs is reported. Thus, BINs with a distance < are seen as identified species, while for those with a distance >3% function as "genetic morpho-species" in the ecological analyses[77]. On this basis, all sequences across all lineages receive a reasonably balanced assignment to taxonomic units (and/or interim species identifications such as the BIN) and information on ecological properties, e.g. pollination[78], is provided at different levels (species, genus, family).

The dataset was then filtered to include only South American records, and in the following way: (1) records were kept that contained, in field 55 ("country"), the South American country names: Argentina, Bolivia, Brazil, Chile, Colombia, Ecuador, Falkland Islands, French Guiana, Guiana, Paraguay, Peru, Suriname, Uruguay, and Venezuela; (2) records were additionally kept if their latitude (field 47, "lat") was between -58.4 and 17 and their longitude (field 48, "lon") was between −85.8 and −30.3. These values were found by taking the extreme north (Punta Gallinas), south (Cape Forward), east (Ponta do Seixas), and west (Punta Parinas) points of the continent. As a buffer, 500 km were added due north, south, east, and west, respectively of those geographic points using the "Measure on Map" function of SunEarth-Tools.com. It was then noted that a large part of the dataset, thus filtered, held also records from several Central American countries, in particular Costa Rica, whose biodiversity on BOLD dwarfs all other South American countries. Thus, a decision was made to additionally include all remaining records from Costa Rica. Finally, a FASTA file annotated with a Process ID (field 1, "processid"), BIN (field 8), taxonomy (fields 10, 12, 14, 16, 18, 20, 22 - "phylum_name", "class_name", "order_name", "family_name", "subfamily_name", "genus_name", "species_name"), geo location data (fields 47, 48, 55), and GenBank ID (field 71, "genbank_accession") was created from the filtered combined TSV file, and then converted into a BLAST database using Geneious v10.2.6 (Biomatters, Auckland, New Zealand). The results were exported and further processed according to methods described by Uhler et al.[77]. Briefly, the resulting csv files, which included the OTU ID, BOLD Process ID, BIN, Hit-%-ID value (percentage of overlap similarity (identical base pairs) of an OTU query sequence with its closest counterpart in the database), Grade-%-ID value (combining query coverage, E-value and identity values for each hit with weights of 0.5, 0.25 and 0.25 respectively, allowing determination of the longest, highest-identity hits), the length of the top BLAST hit sequence, as well as the phylum, class, order, family, genus and species information for each detected OTU were exported from Geneious and combined with the OTU table generated by the bioinformatic pre-processing pipeline. As an additional measure of control other than BLAST, the OTUs were classified into taxa using the Ribosomal Database Project (RDP) naïve Bayesian classifier[79] trained on a cleaned COI dataset of Arthropods and Chordates (plus outgroups; see Porter & Hajibabei[80]). OTUs were also annotated with the taxonomic information from the NCBI (downloaded from "https://ftp.ncbi.nlm.nih.gov/pub/taxonomy/"), followed by the creation of a taxonomic consensus between BOLD, NCBI and RDP.

By restricting the library to BINs with georeferenced records from Central and South America, we limit the references in such a way that, for example, a newly introduced species from New Zealand would not be recognized without a reference from South America. Conversely, blasts against the global database showed that a vast number of species identifications appear, which is completely implausible and complicates further ecological interpretations. The final data set of arthropods comprised 4557 BINs from 24 orders dominated by Lepidoptera (40%) followed by Diptera (32 %), Hemiptera (10%), Hymenoptera (7%), Coleoptera (6%) and Trichoptera (2%).

## Statistical analyses

All statistical analyses were performed in R v. 3.6.3 (Sound indices on high performance cluster) and 4.1.2 (all other analyses)[81]. We first calculated the richness of vocalizing vertebrates per plot and the richness of species observed in the old-growth forest plots as two measures of richness. We then applied non-metric multidimensional scaling ordination (NMDS) using the function metaMDS in the package "vegan"[82] with the Bray-Curtis distance to the community data. Here the function applies a principal component analysis (PCA) on final NMDS values in order to rotate the resulting axes such that the first axis accounts for maximum variance (first principal component). In this way, a major

first and second axis can be calculated as in other ordinations. The score of the first axis was extracted as baseline for further modelling. As this axis revealed a linear gradient of community recovery (Fig. 2a), we generally hypothesized a similar relationship for other indicators, too, and applied linear models in subsequent analyses. The same technique was applied to the community of bird species derived from the CNN analyses as well as to the insect communities identified by metabarcoding.

We finally modelled the axis scores and the two richness values of the vocalizing vertebrate communities, as well as the first axis of nocturnal insect communities with the acoustic indices, which were averaged over the whole two-week period using all sound files collected, and the first axis of CNN-based bird communities using a linear Gaussian model. We used the same model for richness values after log transformation to obtain comparable adjusted $R^2$ for all models. To check the residuals of our models for spatial independence, we used cross-correlograms provided in the package "ncf"[83]. These showed spatial independence of the residuals in all six models (Fig. S4). To test the difference in predictions of Community axis scores by acoustic indices and observed scores based on expert identification we used a linear mixed model and a recovery category specific estimate of the method with plot as random effect to take into account that two observations were from the same plot.

### Reporting summary
Further information on research design is available in the Nature Portfolio Reporting Summary linked to this article.

### Data availability
Raw data for all analyses generated in this study are publicly available from Figshare (https://doi.org/10.6084/m9.figshare.23620323). For the review process, data are available from Figshare (https://figshare.com/s/6db91d2e9b1efd13215b). The FASTQ raw files for metabarcoding of nocturnal insects generated in this study as well as BIN-Plot community data with additional information on BINs by BOLD are publicly available from Dryad (https://doi.org/10.5061/dryad.59zw3r2dm). ImageNet (https://www.image-net.org/) was used to pre-train convolutional neural network for AI bird identification. Insect sequences from light trapping were blasted against GenBank (ftp://ftp.ncbi.nlm.nih.gov/blast/db/) and BOLD (www.boldsystems.org).

### Code availability
Annotated R code, including the data needed to reproduce the statistical analyses and figures, is publicly available from Figshare (https://doi.org/10.6084/m9.figshare.23620323).

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

## Acknowledgements

This work was supported by the Deutsche Forschungsgemeinschaft (DFG). J.M., H.M.S., N.B., H.F., N.G., A.F-L. and D.A.D. here received funding via the DFG Research Unit REASSEMBLY (FOR 5207). Z.B. was supported by the Prince Albert II of Monaco Foundation, grant 3386. We thank the Fundación Jocotoco and Fundación Tesoro Escondido for logistic support and permission to do research on their reserves. We especially acknowledge local support from the staff: Katrin Krauth (manager of the Chocó Lab); Bryan Tamayo (plot manager); Lady Condoy, Leonardo de la Cruz, Franklin Quintero, Jefferson Tacuri, Jordy Ninabanda, Sílvia Vélez, Ismael Castellano, Fredi Cedeño (para-biologists); Alcides Zambrano (Canandé reserve staff); Citlalli Morelos-Juarez, Yadira Giler, Patricio Encarnacion, Ariel Villigu, Patricio Paredes and Adriana Argoti (Tesoro Escondido reserve staff).

## Author contributions

J.M. and H.M.S. perceived the idea for this manuscript. J.M., H.M.S., N.B. and C.J.T. designed the concept of the study. J.M. analyzed the data and wrote the first manuscript draft and finalized the manuscript. Je.M. conducted the metabarcoding. H.M.S., Z.B. and S.S. considerably supported the writing of the manuscript and advised on analyses. J.M., A.B., S.S., D.R., P.K., D.B., H.F., N.G., A.F., D.A.D. conducted the field work. R.G., J.F. and A.A. identified the species. O.M. analyzed the sound indices. G.A.L., T.N.d.M., J.L., M.C-C conducted the convolutional network analyses. All authors commented on the manuscript.

## Funding

## Competing interests

The authors declare no competing interests.

## Additional information

¹Field Station Fabrikschleichach, Department of Animal Ecology and Tropical Biology, Biocenter, University of Würzburg, Glashüttenstr. 5, 96181 Rauhenebrach, Germany. ²Bavarian Forest National Park, Freyungerstr. 2, 94481 Grafenau, Germany. ³Fundación Jocotoco, Valladolid N24-414 y Luis Cordero, Quito, Ecuador. ⁴Technical University of Munich, School of Life Sciences, Ecosystem Dynamics and Forest Management Research Group, Hans-Carl-von-Carlowitz-Platz 2, 85354 Freising, Germany. ⁵Berchtesgaden National Park, Doktorberg 6, Berchtesgaden 83471, Germany. ⁶Saxon-Switzerland National Park, An der Elbe 4, 01814 Bad Schandau, Germany. ⁷Yanayacu Research Center, Cosanga, Ecuador. ⁸Biodiversity Field Lab (BioFL), Khamai Foundation, Quito, Ecuador. ⁹Pasaje El Moro E4-216 y Norberto Salazar, EC 170902 Tumbaco, DMQ, Ecuador. ¹⁰Rainforest Connection, Science Department, 440 Cobia Drive, Suite 1902, Katy, TX 77494, USA. ¹¹Ecological Networks Lab, Department of Biology, Technische Universität Darmstadt, Schnittspahnstr. 3, 64287 Darmstadt, Germany. ¹²Phyletisches Museum, Institute for Zoology and Evolutionary Research, Friedrich-Schiller-University Jena, Jena, Germany. ¹³Animal Population Ecology, Bayreuth Center for Ecology and Environmental Research (BayCEER), University of Bayreuth, 95440 Bayreuth, Germany. ¹⁴Grupo de Investigación en Biodiversidad, Medio Ambiente y Salud-BIOMAS-Universidad de las Américas, Quito, Ecuador. ¹⁵Departamento de Biología, Facultad de Ciencias, Escuela Politécnica Nacional, Av. Ladrón de Guevara E11-253, CP 17-01-2759 Quito, Ecuador. ¹⁶AIM - Advanced Identification Methods GmbH, Niemeyerstr. 1, 04179 Leipzig, Germany. ¹⁷University of Wisconsin-Madison, Department of Forest and Wildlife Ecology and The Nelson Institute for Environmental Studies, 1630 Linden Drive, Madison, WI 53706, USA. ✉e-mail: Joerg.Mueller@npv-bw.bayern.de

