## [Peer Review File · Nature Communications]

Reviewers comments in Nature and replies

Referee #1 (Remarks to the Author):

1. This manuscript brings together multiple types of evidence to evaluate and compare acoustic approaches for evaluating forest regeneration. Working in regeneration sites of different ages and land use history in Ecuador, the authors use manual annotation of soundscapes, acoustic indices, and machine learning approaches to assess whether acoustics reflects the regeneration stage of the forest. In addition, they compare the acoustic metrics against nocturnal insect DNA barcode data, using the nocturnal insect community as an independent (non-acoustic) metric of forest regeneration. The authors find correlations between the regeneration of the forest (age or insects) and the acoustic metrics. The association between regeneration stage and acoustics is particularly strong with the community composition metrics in the expert annotated data and the machine learning identifications, and weaker (though present) in the acoustic indices.

This is a timely topic. There is widespread interest in using acoustics to evaluate habitats, for conservation and particularly as it pertains to carbon markets. Other researchers have investigated the relationship between acoustics and forest regeneration, but generally in situations with fewer replicates, fewer taxonomic groups, shorter timescale of regeneration, and less-controlled experimental designs. In addition, these authors evaluate multiple approaches on the same dataset, making it easy to evaluate the relative strengths of each approach. This manuscript and the findings represent a substantial amount of coordinated effort across a variety of domains, from detailed taxonomic expertise through computational and genetic approaches.

#Reply: We thank for this positive evaluation and are glad that the reviewer finds the unique combination of methods valuable.

2. The presentation of the visuals left some room for improvement in the review PDF (although this may not be a feature of the original figures). Several of the figures have a stray gray line at the edge or are cut off (e.g. Fig 3 where the final value on the x-axis is only partially visible). The visuals are scientifically informative and effective. For a high profile journal where a key figure may be seen by a large number of people, it may be worth investing in the visual appeal and stand-alone interpretability of a few of the core figures that may be viewed in relative isolation from other materials in the paper.

Reply: We thank the reviewer for this comment. We revised Figure 1-3 and Figure S3 to improve quality and also the colours to be suited for colour blind people.

3. Methodologically, it's a creative approach to use the nocturnal insect community as a non-acoustic metric of biodiversity against which acoustic metrics can be compared. As someone who studies acoustics of nocturnal tropical forest insects, this makes me cringe just a bit. A substantial portion of the nighttime soundscape is comprised of insect sound (especially crickets at the audible frequencies, but katydids at high frequencies as well). Looking in detail at the insects represented in the night insect sample, the majority appear to be Lepidoptera and Diptera (as I would anticipate with this structure of light trap). These taxa are not entirely silent, but they do indeed make minimal contributions to the soundscape, so the actual inference should be reasonably solid. However, it might make sense to restrict the insect reference set to these two major groups (or exclude crickets and katydids) to a) minimize the chance that night-calling orthopterans are represented both in the

acoustics and in the DNA and b) to prevent confusion in people who think of insects as a dominant component of tropical nocturnal soundscapes.

Reply: We thank the reviewer for this comment and we agree that it's important to distinguish between the sound-making and relatively quiet insect groups. In the revised manuscript, we have rerun the analyses after excluding the 13 species that would likely be also heard. Reassuringly, the axes of both approaches were strongly correlated (>0.99). We added this finding to the revised text. It reads now (Line 207-209): "In further testing, we excluded the 13 vocalizing insect species present in our light traps, which did not change the axis of species composition (Pearson correlation $\rho=0.986$), indicating that light trap assemblages are predominantly built by non-vocalizing insects."

General comments

4. In introduction, indices are explained and hypotheses developed and then AI approaches appear without much warning. I might restructure this section a bit to say that both approaches are considered and then address each. Right now, these feel like two different threads that sort of alternate turns rather than come together in a single introduction.

Reply: We have restructured this part of the text, introducing both approaches equally and including hypotheses (L102-116): "In tandem with the advancement of soundscape indices derived from audio files, techniques for discerning specific animal species from soundscape recordings are also evolving³⁵. More recently, artificial intelligence (AI) models, such as deep Convolutional Neural Networks (CNN), have been developed to identify birds, bats or amphibians³⁶⁻⁴⁰. When such models become available for many species in a region, species communities could be determined automatically, which has however not been sufficiently investigated to date. A major bottleneck for machine learning approaches, including CNNs, is the need of large training datasets. This is particularly challenging in the hyper-diverse communities of tropical forests⁴¹. Yet, it is increasingly feasible to identify entire vocalizing communities to species level due to the accumulation of labelled acoustic datasets and progress in the development of tools (e.g. Arbimon, BirdNET). Hence, the combination of innovative methods may allow encompassing biodiversity assessments across multiple sites, leveraging community, functional, and phylogenetic diversity measures. As community composition has been shown to predict environmental gradients^{42,43}, we hypothesize that community composition derived from AI will be a powerful surrogate for recovery gradients in tropical forests."

5. > restoring tropical forests is key to combatting the twin crises of our times, biodiversity loss and climate change

I would be a bit circumspect about identifying these as the two crises of our time - humanitarian issues likely loom large in areas beyond biology.

Reply: We agree and rephrased the sentence as (L58-59): "As such, restoring tropical forests is key to counteract two of the major crises of our times, biodiversity loss and climate change"

6. The authors use time as a proxy for how thoroughly a forest has recovered. It may be valuable to spend a sentence on why it is not always possible to just use time as a measure of recovery (rather than the more complicated assessment of biodiversity). For example, time alone may not be reflective of regeneration because of active reforestation efforts, ongoing economic/logging/hunting activities, etc).

Reply: We thank the reviewer for this comment. We fully agree and added a sentence as recommended. It reads now (L155-163): "The positive correlation between recovery time and

community composition may raise the question of why not just use time elapsed as a proxy for biodiversity recovery, such as sometimes done with carbon accumulation¹³. We found a substantial overlap of species compositions between early and late recovery stages, demonstrating the high variability of the different pathways to recovery, even within one landscape. This supports recent findings that showed important impacts of forest cover at the landscape scale on predictability of forest regeneration⁴⁷. Importantly, additional anthropogenic impacts as logging or hunting^{20,48} can strongly affect the local fauna, but are not represented by recovery age. “

7. >In our dataset, we found more species and more vocalizations in agricultural plots

Another possible hypothesis for high diversity is that more species transit through these open sites (but are not necessarily resident and using the habitat)

Reply: We agree and added this as a caveat to the finding (L228-234): “More species might transit through these open sites, or this might be due in part to a higher vertical structure complexity of old-growth forests: our recorders were always placed about 2 meter above ground, possibly underestimating canopy species particularly in old-growth plots with very large trees. Additionally, denser vegetation in old-growth forests may limit sound propagation⁵⁴. However, experimental bird call playback experiments in other countries revealed similar patterns for different recovery stages in tropical forests, which supports similar probabilities of recording⁴⁸.”

8. >We first investigated how well the total vocalizing vertebrate species community identified by experts represents the forest recovery gradient.

At some point in the paper, I would acknowledge that this ‘vocalizing vertebrate’ metric does not include bats, which are a major chunk of tropical vertebrate diversity. I think that this is totally fine in the analysis and study design, but an important distinction to keep in mind when drawing broader conclusions about ‘vocalizing vertebrates’ in a forest restoration context.

Reply: Great point; we fully agree and revised the text accordingly (L122-127): “We carried out i) manual, expert-led identification of vocalizing birds (183 species), mammals (3 species, excluding bats) and amphibians (41 species) within simultaneous time windows covering 28 minutes in all plots across two days, ii) a soundscape index analysis for two weeks of recordings including the days analysed by the experts, and iii) for the same time span a presence and absence of 75 selected bird species identified with a CNN model, trained on an independent dataset.”

9. I would have identified the following paper as potentially relevant uncited literature, but given that the lead author is also an author on this manuscript, I leave it to the discretion of the authors whether to include it:

Campos-Cerqueira, M., Mena, J. L., Tejada-Gómez, V., Aguilar-Amuchastegui, N., Gutierrez, N., & Aide, T. M. (2020). How does FSC forest certification affect the acoustically active fauna in Madre de Dios, Peru?. *Remote Sensing in Ecology and Conservation*, 6(3), 274-285.

Reply: Thank you for this suggestion, we have now included this citation.

Specific comments

10. I’m a bit confused by the paragraph that begins on line 223. Does this refer to a figure? If not, where are the polygons that are referenced in the last line (229-230)?

Reply: Thank you for spotting this error. We have removed the last sentence.

11. In extended figure 3, the red and green will be indistinguishable for colorblind readers, although the information conveyed by the colors here is pretty easily inferred even without the color.

Reply: We have revised the colour selection using a colour blind set. The Figure is now Fig S3.

12. Line 520 “Here internal a principal component analysis (PCA) was applied to the NMDS”. Is the word ‘internal’ supposed to appear here?

Reply: In the revised manuscript, we clarify that we applied the nmds function in vegan, which has implemented a PCA function on the results, putting the NMDS results to major axis (rotation with the highest amount of variation in axes 1 and 2). In this way, they can be interpreted as first major axis, second major axis and so on as in other ordinations (L404-408): “We then applied non-metric multidimensional scaling ordination (NMDS) using the function metaMDS in the package vegan62 with the Bray-Curtis distance to the community data. Here the function applies a principal component analysis (PCA) on final NMDS values in order to rotate the resulting axes such that the first axis accounts for maximum variance (first principal component). In this way a major first and second axis can be calculated as in other ordinations.”

Laurel Symes

Reply: Thank you Laurel for your very helpful comments. We added you to the acknowledgement.

Referee #2 (Remarks to the Author):

13. This paper demonstrates that CNN recognisers, even of only some species, and acoustic indices, can detect differences in species composition among tropical forest plots with different periods of time since disturbance. It is a large piece of work with many replicate plots, and does not suffer from the problems of spatial autocorrelation from which many such studies suffer.

The work is novel in scope (many sites), the comparison of different levels of impact is not done very often. In addition, the use of AI recognisers as well as combinations of indices in the same paper is novel. Using combinations of indices has been done in several works, but not together with other means of examining biodiversity.

I would judge that the data & methodology, validity of approach, quality of data, quality of presentation are good.

Reply: We thank the reviewer for this positive response, and we are glad that the reviewer found our analysis valuable.

14. The use of statistics is mostly good, but I wonder if it is appropriate to use nMDS 'axes' as a variable in a regression. I had heard that it was a problem, and when I looked it up, I found this advice: "nMDS plots don't technically have axes. Plots can be enlarged, reduced, rotated, flipped arbitrarily, and all that matters are the relative distances among objects in the chosen number of dimension in which the ordination has been produced. OK, axes (and scores) are used to plot the result, but they have no interpretation. What you want, to do what you describe, is output from an ordination method that explicitly tries to fit axes in the full multivariate space. PCA does this, and the axes are interpretable. PCoA is another option which allows a wider range of resemblance measures to be used. Finally, mMDS (metric MDS) might be an option, as the axes are interpretable, but mMDS

often has high stress owing to the way it works, so something like PCA/PCoA might be better." So there may need to be some reanalysis there. I think in spite of the above, the conclusions make sense and seem to be what we expect. I guess some people would be surprised by the declining species diversity with recovery from disturbance, but we have also found high diversity in sound and species in disturbed areas as well, often because of a sort of ecotone effect.

Reply: We thank the reviewer for this careful comment. The function we used takes these concerns into account. We explain this better in the revised manuscript. Please also see our response to comment 12 to see the new next.

15. A recent paper using combinations of indices to predict biodiversity at a range of sites is Allen-Ankins, Slade, Donald T. McKnight, Eric J. Nordberg, Sebastian Hoefer, Paul Roe, David M. Watson, Paul G. McDonald, Richard A. Fuller, and Lin Schwarzkopf. 2023. Effectiveness of acoustic indices as indicators of vertebrate biodiversity. *Ecological Indicators* 147: 109937. This may have come out since this paper was submitted.

Reply: We thank the reviewer for this recommendation, which fits well to our manuscript. We added this to our manuscript.

16. The study is clearly presented, and clear, I could understand what they had done and why.

Reply: We thank the reviewer for the constructive comments.

Referee #3 (Remarks to the Author):

17. Please note, as requested by the editor, my review below focusses on the soundscape CNN analysis components, and statistical analyses more generally in this study. My expertise does not extend to tropical forest restoration ecology or meta-barcoding survey methods, and therefore I am unable to comment on those aspects of the manuscript.

Reply: We thank the reviewer for the evaluation, which helped to improve our manuscript.

Whilst the issue of tracking biodiversity scalably within restoration projects is certainly worth addressing, and I applaud the scale of work covered in this study, I had many concerns with the manuscript which I have detailed below.

Major

18. My first major concern is regarding the validation of the CNN for vocalisation detection. Whilst the supplementary information details how validation was performed with a batch of 200 1-minute files (L583) only metrics for mean F1, precision, and recall are reported (L586-588). However, the training samples per class were highly variable, and thus the classifier is highly likely to be biased in its outputs. Using a fixed threshold across a classifier with such highly imbalanced classes is likely to bias metrics towards the classes with most numerous samples, and there is no evaluation of the precision/recall of the classifier by sound class. I imagine the accuracy metrics for the most common species with a large number of training samples will be very good (hence the decent mean F1/precision/recall) but there may be many species where accuracy is pretty poor – and these should be discounted from further analysis.

Reply: We agree with the reviewer that the number of training samples per class can significantly impact the performance and behavior of a classifier. Species' rarity is a common phenomenon in

tropical biomes, and we have used different techniques to mitigate this bias, such as data augmentation and resampling, which we have detailed in the revised manuscript. In addition, model evaluation metrics were computed for each species and averaged, and we have found no significant pattern in the relationship between the number of training data and precision and recall per class. We have added Table S1 to display additional details about model performance and evaluation for individual species. In the revised manuscript, we included new text describing the model evaluation process. (L483-494): “The model was finally evaluated against a dataset including the validation set and an additional 200 expert-labelled recordings from similar habitats as in our study (Table S1). The mean average-precision across the classes of the best model was 86.9%. This score represents the average precision across all thresholds, using the changes in recall between each consecutive threshold as weights for the average. At an output score threshold of 0.5, the model achieved a mean F1-score across classes of 77.2%, a mean precision of 82.9% and a mean recall of 80.7%. For a threshold of 0.8, the mean F1-score, precision, and recall were 78.9%, 89.4%, and 77.5%, respectively. The scores are averaged across species to balance the contribution from each species and avoid bias for the most common species. S2 contains species-specific scores for each of these metrics. There is evidently no strong bias toward the most common classes (Pearson correlation $\rho=0.07$, $p=0.48$, Precision 0.8 $\sim \log(\text{Presences})$).” Furthermore, we have added additional clarification within the primary manuscript, elucidating that the artificial intelligence (AI) model used for avian species identification underwent development and training using data and species exclusively from the geographic region of our study and that this model training was conducted before the commencement of the sampling procedures outlined in the study.

19. Details of audio recording devices were given in the supplementary (though I think these at least deserve to be in the methods). Two recorders were used in this study, with vastly differing mounting heights (1.5m vs in the canopy) and recording frequencies (48kHz vs 12kHz). More details on my concerns here are given in the line-by-line comments, but this could be a fairly major bias in your dataset that affects all audio-derived results.

Reply: We acknowledge the concerns raised by the reviewer and would like to address them accordingly.

First, we note that the two different devices were used only in developing the AI-model. The study itself, and hence also its conclusions, are based on a single system always positioned at the same height above ground. Thus, there is no major bias that would affect all audio-derived results. We believe that our study rather shows that it is robust against potential biases caused by using two systems, as such biases should reduce the accuracy of the CNN model rather than enhancing it. In the revised manuscript, we explain this better (L122-127): “To pioneer the assessments of biodiversity recovery in tropical forests, we sampled soundscapes simultaneously with the same devices and protocol in 43 plots along a recovery gradient as a space for time substitute approach (Figure S1).”

The two different devices were used during the model training, in order to make the model more generalizable. To mitigate the aforementioned bias, we made substantial efforts to include data from devices deployed in the understory, employing a 48 kHz sampling rate, during both the training (14%) and evaluation (23%) stages of the Convolutional Neural Network (CNN) model. Furthermore, the CNN model was applied to generate predictions on raw recordings obtained from devices configured similarly to the AudioMoths, which were deployed in the understory and operated at a sampling rate of 48 kHz.

20. My final major question is on the generalisability of the soundscape approaches. As seen in the literature, the authors here report varying relationships between their soundscape indices and expert-derived species communities. Some indices increase with species richness, others decrease, others have no relation. Without collecting the expert observational data or the insect meta-barcoding, how can someone setting up a new restoration project know which soundscape indices to use, how they will respond to their gradient, and whether it will show any signal at all? This really limits the applicability of the approach, as this study only covers one area, and so unless a new study is established in this corner of Ecuador, it seems unlikely that the soundscape results presented here will change our best understanding of how to use soundscapes to monitor biodiversity.

Reply: We agree with the reviewer that currently, it is too early to draw generalized, global conclusions on the usefulness of acoustic indices in estimating species recovery. However, by comparing the CNN model to detailed ecological data, we propose that such models can be developed in the coming years. Our study brings an important piece of evidence, documenting the behavior of the different indices with expert-validated biodiversity metrics. We have highlighted this caveat in the discussion (L273-277): “To generalise the soundscape approaches, AI-models for vocalizing animals have to be improved globally and their application to environmental gradients as the recovery gradient in our study has to be validated regionally. Therefore, we urge the conservation community to prioritize the creation of global sound repositories for taxa beyond birds, based on which machine learning models can be rapidly improved and extended.”

21. L74: “avoid greenwashing” seems overly non-specific – not sure that a lack of monitoring is the only barrier in the way of greenwashing

Reply: We agree and revised the text. It reads now (L76-77): “and one important step to avoid greenwashing”.

Furthermore we added a sentence to the conclusions (L282-284): “This, in combination with open access strategy, could also help to reduce greenwashing in nature conservation.”

22. L83: “forest loss and degradation can be tracked by using soundscapes”

This is a very concrete statement, which is then directly contradicted in the following paragraph which discusses the inconsistencies of soundscape indices – and their inability in fact to reliably track changes in biodiversity. If the authors are referring to how vocalization detection can be used, I’d recommend keeping terminology separate so people don’t assume “soundscapes” = “soundscape indices”.

Reply: We thank the reviewer for the comment. In the revised manuscript, we have made the distinction clear between soundscapes (which can be used in various analyses) and soundscape indices.

23. L91: I’d suggest the below citation which gives more comprehensive evidence to the author’s claims: Is there an accurate and generalisable way to use soundscapes to monitor biodiversity?
<https://www.biorxiv.org/content/10.1101/2022.12.19.521085v1>

Reply: We thank the reviewer for the comment and we have added the paper to our manuscript.

24. L100: Now the introduction returns to vocalisation detection approaches using machine learning. This is difficult to follow and I'd suggest a restructure of the entire introduction so vocalisation detection and soundscape analysis approaches are clearly delineated

Reply: We have rephrased that part of the introduction as follows (L102-116): "Apart from soundscape indices, techniques for discerning specific animal species from soundscape recordings are also being developed³⁵. More recently, artificial intelligence (AI) models, such as deep Convolutional Neural Networks (CNN), have been developed to identify birds, bats or amphibians³⁶⁻⁴⁰. When such models become available for many species in a region, species communities could be determined automatically, which has however not been sufficiently investigated to date. A major bottleneck for machine learning approaches, including CNNs, is the need of large training datasets. This is particularly challenging in the hyper-diverse communities of tropical forests⁴¹. Yet, it is increasingly feasible to identify entire vocalizing communities to species level due to the accumulation of labelled acoustic datasets and progress in the development of tools (e.g. Arbimon, BirdNET^{42,43}). Hence, the combination of innovative methods may allow encompassing biodiversity assessments across multiple sites, leveraging community, functional, and phylogenetic diversity measures. As community composition has been shown to predict environmental gradients^{44,45}, we hypothesize that community composition derived from AI will be correlated with recovery gradients in tropical forests."

25. L106: Why is it becoming increasingly feasible?

Reply: We revised the paragraph, please see reply to comment 24 above.

26. L110: So this is using a space for time substitution? I'd explicitly state that – rather than the assumption which would be that it's a longitudinal study

Reply: In the revised manuscript, we added this information (L117-119): "To pioneer the assessments of biodiversity recovery in tropical forests, we sampled soundscapes simultaneously in 43 plots along a recovery gradient as a space for time substitute approach (Figure S1)."

27. Fig 1a: Is this data based on the expert identifications or the CNN model? Should be explicitly stated either in the figure itself or the caption.

Reply: In the revised manuscript, we added the information. It is based on the full expert identification.

28. Fig 1a: The text "Oldgr" "Pregll" etc is too small to read without zooming in

Reply: We have increased the size of this text in the revised figures.

29. L143: "Old growth forests were distinguishable": is there some statistical analysis to back up this claim? If not, it should be made explicit this is a qualitative statement based on the 2D PCA. Also, perhaps the PCA components 3/4/5/etc would have separated the other classes of plot – has this been tested? If not, what is the motivation (beyond visualisation purposes) for only using 2 PCA dimensions of the species community?

Reply: We thank the reviewer for the comment. We have revised the text to clarify this. It reads now (L149-151): “Most old-growth forests were rather distinguishable in community composition from others along the first ordination axis, with the exception of one pasture regenerating for 34 years (Fig. 1a). This is in contrast to bird communities in tropical dry forests with earlier broader overlap with reference sites⁴³”

30. L166-L172: This four sentence explanation for how soundscape indices may have saturated is overly speculative in my opinion. To make such a point in the heart of the results, the authors should be presenting real data which supports their hypothesis, and not entirely relying on invoking the acoustic niche hypothesis, which itself is highly debated (e.g., see evidence from reference 26 in the introduction).

Reply: To bolster our discussion here we added new references. We have also rephrased the sentence to indicate the speculative nature of this explanation (L180-181): “The inconsistency in old-growth forests could be partially because soundscape indices may saturate at high diversity values^{16,50}.”

31. L173: Before introducing results from the CNN, it would be helpful to have a brief explanation for how this novel bird vocalisation detection CNN was developed. Did the authors collect data specifically for this project, or was it harvested from existing libraries? What motivated the choice of the 25% of species (L177)? Was it transfer learned or a completely new model? I’m sure some answers will be in the methods, but this context is essential for assessing the applicability of the results to others interested in using the approach and should be covered briefly in the main text.

Reply: In the revised manuscript, we added two sentences for clarification (L186-189). It now reads: “In the next step, we applied an independent AI model for bird species identification, developed and trained in the region of our study prior to our sampling. Despite the model being able to identify only ~25% of species detected by the experts in our data, the model-derived first community axis was the single best predictor for expert-derived community composition.”

32. Fig 2: Perhaps a panel that would add a significant amount of trust in the results here would be to measure and display explicitly the precision per species of the CNN model.

Reply: We have added an additional explanation (L486-497) and a table (Table S1) about the CNN development in the methods and supplementary materials: “The model was finally evaluated against a dataset including the validation set and an additional 200 expert-labelled recordings from similar habitats as in our study (Table S1). The mean average-precision across the classes of the best model was 86.9%. This score represents the average precision across all thresholds, using the changes in recall between each consecutive threshold as weights for the average. At an output score threshold of 0.5, the model achieved a mean F1-score across classes of 77.2%, a mean precision of 82.9% and a mean recall of 80.7%. For a threshold of 0.8, the mean F1-score, precision, and recall were 78.9%, 89.4%, and 77.5%, respectively. The scores are averaged across species to balance the contribution from each species and avoid bias for the most common species. S2 contains species-specific scores for each of these metrics. There is evidently no strong bias toward the most common classes (Pearson correlation $\rho=0.07$, $p=0.48$, Precision $0.8 \sim \log(\text{Presences})$).”

33. L213-218: “may indicate that species in more mature forests experience....” As before, there is a lot of speculation in this results paragraph which is almost fully based on the acoustic niche hypothesis. This should either be supported by novel data or moved to a speculative discussion section (independent from the results).

Reply: The submission format for presentation in Nature required us to combine the results and discussion into one section. We have now revised the possible explanation to rely less on the acoustic niche hypothesis (L240-242): “The increase of Soundscape Saturation along the recovery gradient (Figure 3) could indicate that more mature forests have an acoustically more diverse communities, resulting in a broader coverage of the frequency spectrum.”

34. L225: “increased with the Entropy” – is this temporal or frequency entropy?

Reply: In the revised manuscript, we clarified that we mean entropy of frequency.

Methods

35. L471: Why was a model pre-trained on ImageNet chosen rather than one pre-trained on the audio specific AudioSet dataset (e.g., VGGish, YAMNET)?

Reply: Previous work at Rainforest Connection has found that models pre-trained on ImageNet outperformed those on AudioSet in certain cases, particularly for complex biological signals. We believe the narrow input width (1 second) and relatively low resolution of the input spectrograms (64 x 96 vs for AudioSet vs. up to 224 x 224 for ImageNet) are the main factors impacting performance in these cases. Other published CNNs of animal sounds have also been pre-trained on ImageNet (e.g. Kahl et al. 2021, Sun et al. 2022). We have explained this point in the revised manuscript including a new reference (L371-374): “While pre-trained models for audio recognition have recently been proposed (YAMNet, VGGish60), they are trained on sounds of limited time duration and resolution (rather than on images of sounds), which can impact their applicability to noisy and complex signals such as found in biodiverse soundscapes.”

Supplementary

36. L544: Please give more detailed information on microphones used in these devices and sensitivities. Also Audiomoths were mounted at 1.5m, whilst Guardians were in the canopy (I imagine 20m+?) Knowing which recorders were used where is important – perhaps this biased results? I imagine across the restoration gradient the canopy height would increase, so in the worst case the soundscape indices could just be picking up on vertical stratification in soundscapes rather than any signal from changing species communities?

Reply: We have added additional information on the hardware used in data collection. Please also see our reply to comment #19: the model was (already) trained on a former dataset separate to the one used for the main ecological analysis presented in this manuscript. Only the former CNN model training and testing data contained a mixture of device configurations, which we believe would have improved the ability of the model to generalize. The data collected for the main ecological analysis here used the same device and configuration (near ground) for each deployment to avoid potentially confounding variables. We have provided additional information

about device deployment in the supplementary material (L444-452): “The algorithm we used was developed independently from this study⁴³. Data used to train the CNN model came from 401,685 1-minute soundscape recordings previously collected in a separate study from 55 sites in and around Mashpi Rainforest Biodiversity Reserve and Canandé Reserve in the Ecuadorian Chocó Forest (500 to 2300 m a.s.l.). Recordings for training were collected between January and December 2019, using Audimoth devices⁶⁶ and Guardian recorders (<https://rfcx.org/guardian>). Audimoths were placed on a tree at the height of 1.5 m and programmed to record 1 min of audio every 10 min for a total of 144 recordings per day at a sampling rate of 48 kHz. Guardian recorders were deployed at the canopy of trees at a height of ~30 m and programmed to record continuously at a sampling rate of 12 kHz.”

37. This is actually referred to in L209, but there it is claimed that recorders were at a lower level in the old-growth forests. Experimental playbacks are mentioned which may address this issue, but no details of the playbacks are presented in this manuscript or its supplementary which makes it difficult to evaluate this data.

Reply: This is a misunderstanding and we have corrected the phrasing – all recorders were positioned at 2m height above ground. We revised the text here (227-230): “More species might transit through these open sites, or this might be due in part to a higher vertical structure complexity of old-growth forests: our recorders were always placed about 2 meter above ground, possibly underestimating canopy species particularly in old-growth plots with very large trees.” **Additionally, we want to clarify that the playback experiments were done by other studies, not ours. We have not made a clearer distinction between our findings and findings of other studies (L231-233):** “However, bird call playback experiments in other countries revealed similar patterns for different recovery stages in tropical forests, which supports similar probabilities of recording.”

38. L559: Guardians were recording at only 12kHz, whilst Audimoths were at 48kHz. Whilst all data may have been resampled to 16kHz, this doesn’t change the fact that there is no data from the Guardians between 6-8kHz bands. Many birds and insects will vocalise in these bands and so this will affect both the soundscape indices and the CNN vocalisation detection model

Reply: We agree with the reviewer that the Guardians’ recordings may have missed some bird species, vocalizing above 6kHz. However, this was at least partially mitigated by using also training data from Audimoths, which included also higher frequency sounds. All data in this project were collected with the same devices in the same way. The soundscape indices have been calculated on data independent from the Guardians. In the revised manuscript, we have clarified the data provenance for each step of our study.

REVIEWER COMMENTS

Reviewer #1 (Remarks to the Author):

The authors have addressed the major reviewer concerns in the revised manuscript. In particular, the distinction between training data (from multiple recorder types and heights) and experimental data (from a single device and height) was clearly stated and address one the primary reviewer concerns. In addition, the information on the distribution and implications of class imbalance were made clear in a way that helps to address reviewer confusion.

In general, the manuscript appears to have been thoughtfully revised.

[Editor's note: Reviewers 2 and 3 from the first round at Nature were not available]

Reviewer #4 (Remarks to the Author):

Muller et al. present an impressive study demonstrating the capacity of deep learning methods in combination with acoustic indices to assess biodiversity recovery in tropical forests using a space-for-time substitution approach. I commend the authors on an important, well-written, and expansive study, which I did not review previously.

I preface this review with a note that I was specifically asked to evaluate the authors' responses to previous reviewers, with a focus on the response to reviewer 3. While I am not an expert on AI methods like CNNs – having worked mainly with acoustic indices and out-of-the-box species classifiers (e.g. Kaleidoscope Pro) – I find the manuscript to be scientifically sound and generally accessible even to those who are not intimately familiar with deep learning soundscape methods, or with tropical forest ecology.

I feel the authors have done a good job of responding to all previous reviewer comments, and I am satisfied with their response to reviewer 3's points about the CNN methods. In my (minor) comments below, I flag only a couple of instances where I think a slight edit may be useful to better resolve prior reviewer comments.

I had only one main sticking point when reading the manuscript, and it was itself a minor one. A series of comments below focuses on how Figures and Tables using the Community Vertebrates PC Axes are cited. I think confusion can arise when trying to interpret these figures/tables if readers aren't reminded what the community vertebrates axis represents. As presented, currently I found myself thinking that these were miscited because they don't explicitly show comparisons along the recovery trajectory, but then I understood when referring back to Figure 1. Please consider using more reader-friendly labels or at least a quick summary in the legend to remind people to refer back to Fig 1, otherwise readers must search the manuscript again to correctly interpret these later figures. I appreciate that in the interest of accuracy, it's not always ideal to write the PC loadings explicitly on the axes, but I found this PC axis to be named in a way that caused me some confusion when attempting to read the display items, so even just a quick label of "Oldgr" and "Pasture" at the extremes of the axis labels, or some reference to Fig. 1 in the legend should fix this (see specific instances with line numbers below).

I otherwise have only a few minor suggestions as follows:

L79 and response to reviewers – I agree with the previous comment that greenwashing is a little vague / comes as a bit of a surprise here. I think a small link is missing to help clarify how cost-effective and generalisable biodiversity censusing techniques can avoid greenwashing. Personally, I don't see a direct link.

L91 and throughout, and response to reviewers – to further distinguish between soundscapes and soundscape indices, I wonder if replacing all uses of "soundscape index" with "acoustic index" might be worthwhile? Increasingly in the literature I'm seeing these indices referred to as acoustic indices rather than soundscape indices. If changing to acoustic indices, effort should be made to clarify that "AI" is an abbreviation of artificial intelligence rather than acoustic indices, though. Perhaps the authors prefer to avoid abbreviation ("AI" is only used 4 times in the manuscript) or stick with soundscape indices, but I thought I'd suggest an option regardless.

I note here that from L177 onwards the authors refer to "acoustic indices". Whether choosing soundscape indices or acoustic indices, best to be consistent throughout.

Para starting 104 – When introducing artificial intelligence models such as deep learning CNNs, I wonder if it might be worthwhile briefly introducing and contrasting these with out-of-the-box models for species identification, which are typically less data hungry, computationally simpler in design (e.g. relying on MCMC vocal separators), but rely heavily on human input (i.e. "supervised machine learning"). To me, the really exciting thing about the continued successful application of deep learning methods in ecoacoustics is that it increasingly makes these simple and often black box approaches to building species recognisers seem more and more obsolete; open-source deep learning methods tend to be much more flexible and may require fewer person hours to generate well-performing models, so it might be nice to see this point briefly introduced here. As currently cast, this paragraph reads as if the only two options for generating ecological insight from soundscape recordings are acoustic indices and deep learning models like CNNs. To even better position the deep learning approaches as a useful tool, I think a brief 1-2 sentence contrast (probably near the beginning of the paragraph) to these simpler species classification algorithms could strengthen the argument for CNNs here. The rest of the paragraph would then remain largely the same.

L108 – "Species communities" sounds odd to me. Maybe "multispecies communities" or "ecological communities" would be better?

Figure 1b – were any of the pairwise comparisons between expert community and sound prediction significantly different (at any site)? If so, might be worth flagging which one(s). If not, perhaps saying so explicitly in the figure legend will help emphasise that the indices performed well.

L164-166 – I didn't follow the logic that overlap of composition between early and late-stage recovery suggests a variety of recovery trajectories. In fact, to me, it seems to suggest the opposite; that composition recovers rapidly and remains largely unchanged during later stages of recovery, unless I'm missing something? Perhaps worth recasting or expanding this point a little.

L187 – I think "acoustic niches" is a more common term than "soundscape niches". Consider rephrasing?

Figure 2b – is linear regression the best approach here? Just by eyeballing the data, it looks like a quadratic term might improve model fit. Was model selection and comparison used to fit these models

or was there an a priori expectation of linearity? I could not really tell from the Methods section. Same goes for other linear model fits throughout.

L208 – While an R-squared value of 0.39 is indeed something I'd consider to be generally "high explanatory power" in ecology (especially for species rich tropical systems), given the general readership of the journal, I wonder if it might be better to tone this down slightly? Those reading from an AI background may disagree that 0.39 is high explanatory power, for instance.

L213-214 – I'd maybe clarify here that indices seem to be useful surrogates for non-vocalising fauna in *this* system, but I'm not convinced there's enough evidence (from other similar studies) to phrase this as a sweeping statement across ecosystems (as it currently reads).

L216 – "Sound indices" is another competing term used to describe acoustic indices. Please change to acoustic indices (or soundscape indices) for consistency to avoid confusion. Also L264. It might be worth using find and replace for "index" and "indices" to catch all instances of this.

L222 – Significant predictors in what sense? I assume $p = 0.05$, but worth stating explicitly in the table legend.

Table 1 – In the legend please briefly explain what the response variables are, and perhaps say something like "see Methods for explanations of acoustic index calculation", in case for some reason a reader jumps straight to table 1.

L227-229 – when citing figure 3 and table 1 here, it's worth reminding readers how community vertebrate axis 1 relates to recovery time (i.e. habitat type). (same with Table 1 cited in L249).

L229 – Is fig S2 mis-cited here? Perhaps I've missed something.

Fig 3 – as with Fig 2, might be nice to include some measure of goodness-of-fit or similar, here. Also, the community vertebrate axis 1 is not that easy to interpret visually without referring to Fig 1. Might it be worth briefly explaining the loadings on this PC Axis in the legend or as simple labels at the Y axis value extremes?

L243-246 – alternatively, might it be that mature forests have a greater richness or abundance of broadband insects which can saturate soundscapes (e.g. cicadas), rather than necessarily a broader diversity of species across taxa?

L256 – repeated "that", delete one.

L287 – I'm not sure I 100% agree that long-term acoustic data is easy to store. Often this data is very large and required a LOT of data storage. Maybe the point was that it's easy to store compared to type specimens? Or to Winkler/pitfall samples? Something else?

L288 – "open access strategy" is a bit too vague for me to understand exactly what is meant here. Also, as above, and in agreement with a previous reviewer, I don't quite see the link to greenwashing. Please remove or recast to better clarify.

L292-293: Maybe rephrase to "the joint biodiversity and climate crises" or similar. Currently feels an odd formulation.

End of comments. Congratulations on producing a manuscript that is important, interesting, and enjoyable to read.

Sam Ross

Reviewer #5 (Remarks to the Author):

I have reviewed the manuscript titled "Soundscapes and artificial intelligence provide powerful tools to track biodiversity recovery in tropical forests" by Müller and colleague and I am providing some comments focusing on the insect metabarcoding parts, as per the editor' request.

I think that the results presented in the MS are very promising and that it would be extremely useful to add soundscapes and artificial intelligence to the list of tools capable of tracking biodiversity recovery in tropical forests. Therefore, I think the work has the potential to be very significant in the field.

I think that this work generally supports the conclusions and claims reported. Although I have some minor comments and questions that I hope may contribute to improve the overall MS.

While I have not found any flaws in the data analysis, I think the interpretation and conclusions on the insect metabarcoding section can be made clearer. The authors appear to have a very interesting dataset, composed by almost 5000 Bins, which can provide a lot of information. I was a bit disappointed as it seems the author did not really explore this dataset, as they did not provided almost any information on the insect diversity recorded. I think the author could address some of the points below and the MS could be improved substantially, with very little effort.

I understand that insect metabarcoding was a relatively small component of this work, but considering the large volume of works published in the last few years, it would be important this work remains consistent with the standards of this field, by ensuring that the insect identification used here is robust.

I have a couple questions for the authors, since I don't seem to find any answer in the manuscript. I was requested to revise the MS focusing on the topic of insect metabarcoding, however some aspects of the MS were not clear to me but important for my understanding of the sampling and analysis:

- The authors have been sampling insects using a light trap, in particular a trap that has been produced mostly to attract Lepidoptera (LepiLED). The authors should probably state this can be considered a bias, since not all nocturnal insects are equally attracted to light. Perhaps the authors could not collect vocalising insects that were not attracted to light? This would have excluded some crucial taxa from their analysis.

- The authors mention that only 13 species of insects were found to be vocalising nocturnal insects (line 210), over a total of almost 5,000 BINs. By removing these species, both indeces and CNN still had a high explanatory power even for species that are not part of the vocalizing tropical animal community. Can the authors expand on this? How is it possible that soundscape indeces and CNN work even if the insects are non-vocalising? I admit this being entirely outside of my area of expertise, but

if the soundscape indices have strong explanatory power, couldn't it be because some of those >4000 BINs may actually include vocalising insects? Considering our scarce knowledge of the species present in high-biodiversity areas in Central and South America, how can the authors safely exclude all other BINs from being "vocalising insects". Additionally, the authors should explain in the methods, which are the 13 vocalising insects they recorded and subsequently removed from the dataset for some of their analyses.

In addition, I have provided a few corrections/comments I hope may improve the work:

Everywhere in MS. While it is true that there may be variant, the general consensus on the use of the word "metabarcoding" is to write it without hyphen. Please remove all instances of "meta-barcoding".

Line 130. The authors have used the BIN system as a proxy for species identification. While this system can be a very valid approach to assess and compare diversity, unfortunately it doesn't automatically translate to species identification. While the results are not reported in the main text, I doubt the authors have "identified all insects using metabarcoding", since many (most?) of the taxa could probably not be identified to species level. This is normal with insect metabarcoding, especially in area with high level of endemism and diversity. I would suggest the author rephrase the sentence with something like: "...with autonomous light traps and assessed insect diversity using metabarcoding.."

Line 256. Repetition of "that". Remove the second "that". "Our data show that not only nocturnal insects recover quickly."

Line 390: Please, specify which model of LepiLED, since this can impact the number of LED lights on the lamp and therefore the different peaks in the light spectrum.

Line 392. "Collections" should start with a lower case: "collections".

Line 396. Please specify which alcohol. Was it ethanol?

Lines 397-400. The Skelton et al. 2022 reference is a very good work, however it is focused on environmental DNA. Since you are working on insect metabarcoding from full specimens, Elbrecht & Leese as well as Martoni et al. 2022 are going more in depth on the issues linked to reads bias due to biomass, primers and taxonomic similarity. The authors should acknowledge that while size filtering is very good practice, this unfortunately cannot guarantee all insects have been recorded, and that reads bias has been avoided. Even closely related species belonging to the same genus may be differently impacted by the same DNA extraction/amplification/sequencing procedure, generating a difference in read numbers (i.e. a bias), that may result in loss of rare species; see Martoni et al. 2022.

Martoni F, Piper AM, Rodoni BC, Blacket MJ. 2022. Disentangling bias for non-destructive insect metabarcoding. PeerJ 10:e12981 <https://doi.org/10.7717/peerj.12981>

Line 401. Full name of gene (cytochrome oxidase) should be italicised. Gene name should be reported as "Subunit I of the mitochondrial gene cytochrome oxidase".

Lines 406-407. This is the issue I mentioned at line 130 linked to the BINs methodology. The fact that a BIN cluster matches a sequence at 90% means that species could NOT be identified. A 90% match of a short fragment of COI means the species is probably in the same genus, but certainly not the

same species. As I mentioned before, this is not a methodological issue, but the authors should be clear about their results. I don't think it is fair to say that a match at 90% "allow comparison with studies based on morphological determination". I would remove the sentence entirely after "genetic similarity)"

Lines 407-408. Why was the library limited to countries from Central and Southern America? And most importantly, how? This could be a relatively major issue.

If the author limited the reference library to samples collected from Central and Southern America, they may have limited their own identification efficacy. If a species is present both in Argentina and in Australia, but was sequenced only from Australia, have the authors removed the data because the sequence is not from Central and Southern America? Therefore impairing their identification system by removing sequences that could in fact match 100% with their data?

Or did the author only included species known to be present in Central and Southern America? If this is the case, the issue could be minor. However, by doing so they won't be able to identify exotic species (i.e. not from Central and South America) that have been introduced to the area.

Line 408. The authors are reporting BINs from 39 orders. This suggests that they recorded more than just insects. What are the extra orders from? Arachnida? Mites? Would it be better to revise the manuscript and use "arthropods" instead of insects? Would it be possible to include some more information on the other orders? For example, "dominated by Lepidoptera (% of BINS) and Diptera (% of BINS), followed by Order (%), Order (%)."

Line 411. Please reference R every time you use it in a manuscript:
R Core Team (2023). R: A language and environment for statistical computing. R Foundation for Statistical Computing, Vienna, Austria. <https://www.R-project.org/>.

Line 515. Please, change "Next Generation Sequencing" with "High Throughput Sequencing". During the past 5-7 years it has been pointed out how these technologies are not "next generation" anymore, since they have been around since 2007-2008, as the authors have used in the same paragraph.

Point to point

Editor

Comment 1: At this stage I do not have specific guidance to offer other than to pay particular attention to Reviewer 5's question on the reference library (see the comment starting with "Lines 407-408"). In addressing the reviewers' concerns, please note that in Nature Communications you have wider margins to expand the text than in the original Nature format used in the first round; in particular, Introduction, Results and Discussion may be up to 6,000 words in total (though preferably 5,000 or less) and there is no word limit on the Methods section.

Reply: We appreciate the positive evaluation and have addressed all comments, including the highlighted ones (Lines 407-408), with corresponding revisions highlighted in red.

Reviewer #1 (Remarks to the Author):

Comment 2: The authors have addressed the major reviewer concerns in the revised manuscript. In particular, the distinction between training data (from multiple recorder types and heights) and experimental data (from a single device and height) was clearly stated and address one the primary reviewer concerns. In addition, the information on the distribution and implications of class imbalance were made clear in a way that helps to address reviewer confusion. In general, the manuscript appears to have been thoughtfully revised.

Reply: Many thanks for the positive evaluation.

Reviewer #4 (Remarks to the Author):

Comment 3: Müller et al. present an impressive study demonstrating the capacity of deep learning methods in combination with acoustic indices to assess biodiversity recovery in tropical forests using a space-for-time substitution approach. I commend the authors on an important, well-written, and expansive study, which I did not review previously. I preface this review with a note that I was specifically asked to evaluate the authors' responses to previous reviewers, with a focus on the response to reviewer 3. While I am not an expert on AI methods like CNNs – having worked mainly with acoustic indices and out-of-the-box species classifiers (e.g. Kaleidoscope Pro) – I find the manuscript to be scientifically sound and generally accessible even to those who are not intimately familiar with deep learning soundscape methods, or with tropical forest ecology.

Reply: Thanks

Comment 4: I feel the authors have done a good job of responding to all previous reviewer comments, and I am satisfied with their response to reviewer 3's points about the CNN methods. In my (minor) comments below, I flag only a couple of instances where I think a slight edit may be useful to better resolve prior reviewer comments.

Reply: Thanks for improvement of our manuscript

Comment 5: I had only one main sticking point when reading the manuscript, and it was itself a minor one. A series of comments below focuses on how Figures and Tables using the Community Vertebrates PC Axes are cited. I think confusion can arise when trying to interpret these figures/tables if readers aren't reminded what the community vertebrates axis represents. As presented, currently I found myself thinking that these were miscited because they don't explicitly

show comparisons along the recovery trajectory, but then I understood when referring back to Figure 1. Please consider using more reader-friendly labels or at least a quick summary in the legend to remind people to refer back to Fig 1, otherwise readers must search the manuscript again to correctly interpret these later figures. I appreciate that in the interest of accuracy, it's not always ideal to write the PC loadings explicitly on the axes, but I found this PC axis to be named in a way that caused me some confusion when attempting to read the display items, so even just a quick label of "Oldgr" and "Pasture" at the extremes of the axis labels, or some reference to Fig. 1 in the legend should fix this (see specific instances with line numbers below).

Reply: Thank you for your feedback. We have made the necessary improvements to the axes and figure captions as per your suggestions

I otherwise have only a few minor suggestions as follows:

Comment 6: L79 and response to reviewers – I agree with the previous comment that greenwashing is a little vague / comes as a bit of a surprise here. I think a small link is missing to help clarify how cost-effective and generalisable biodiversity censusing techniques can avoid greenwashing. Personally, I don't see a direct link.

Reply: Thank you for your input. We have modified the text to clarify the main message, as suggested. The revised sentence now reads (line 73-81): *"In particular, market-based conservation mechanisms that may rely on forest restoration, such as payments for ecosystem services, biodiversity offsets and credit markets, as well as e.g. forest sustainability certification, urgently require a cost-effective, transparent and generalizable biodiversity measurement and monitoring tool. This need arises to facilitate scalability in alignment with UN targets. Such tool is also fundamental to help prevent greenwashing: without the requirement and tool to monitor biodiversity, carbon-focused actors may plant simple, monoculture plantations, instead of forests that have the potential to become biodiverse and resilient with proper restoration."*

Comment 7: L91 and throughout, and response to reviewers – to further distinguish between soundscapes and soundscape indices, I wonder if replacing all uses of "soundscape index" with "acoustic index" might be worthwhile? Increasingly in the literature I'm seeing these indices referred to as acoustic indices rather than soundscape indices. If changing to acoustic indices, effort should be made to clarify that "AI" is an abbreviation of artificial intelligence rather than acoustic indices, though. Perhaps the authors prefer to avoid abbreviation ("AI" is only used 4 times in the manuscript) or stick with soundscape indices, but I thought I'd suggest an option regardless.

Reply: Great comment. We followed your advice and changed *soundscape index* to *acoustic index*. Furthermore, we removed AI from the text using artificial intelligence to avoid confusion.

Comment 8: I note here that from L177 onwards the authors refer to "acoustic indices". Whether choosing soundscape indices or acoustic indices, best to be consistent throughout.

Reply: We now use only acoustic indices.

Comment 9: Para starting 104 – When introducing artificial intelligence models such as deep learning CNNs, I wonder if it might be worthwhile briefly introducing and contrasting these with out-of-the-box models for species identification, which are typically less data hungry, computationally simpler in design (e.g. relying on MCMC vocal separators), but rely heavily on human input (i.e. "supervised machine learning"). To me, the really exciting thing about the continued successful application of

deep learning methods in ecoacoustics is that it increasingly makes these simple and often black box approaches to building species recognisers seem more and more obsolete; open-source deep learning methods tend to be much more flexible and may require fewer person hours to generate well-performing models, so it might be nice to see this point briefly introduced here. As currently cast, this paragraph reads as if the only two options for generating ecological insight from soundscape recordings are acoustic indices and deep learning models like CNNs. To even better position the deep learning approaches as a useful tool, I think a brief 1-2 sentence contrast (probably near the beginning of the paragraph) to these simpler species classification algorithms could strengthen the argument for CNNs here. The rest of the paragraph would then remain largely the same.

Reply: Great point. We fully agree and revised the text accordingly. It reads now (line 106-114):
Apart from acoustic indices, techniques for discerning specific animal species from soundscape recordings are also being developed. Out-of-the-box models for species identification are typically less data hungry, computationally simpler in design (e.g. relying on MCMC vocal separators), but depend on human-guided feature engineering (i.e. “supervised machine learning”), introducing potential subjectivity that could hinder performance, particularly with diverse or noisy datasets. More recently, artificial intelligence (AI) models, such as deep learning methods as Convolutional Neural Networks (CNN), have been developed to identify birds, bats or amphibians. They tend to be much more flexible and may require fewer person hours to generate well-performing models.

Comment 10: L108 – “Species communities” sounds odd to me. Maybe “multispecies communities” or “ecological communities” would be better?

Reply: The term "Species communities" is a fitting and scientifically well recognized expression to describe the intricate interdependencies and interactions among various species within specific ecological niches. It encapsulates the dynamic relationships, coexistence patterns, and ecological roles that species collectively form within ecosystems. The term has been used back in 1970's and in current research (https://www.researchgate.net/profile/Daniel-Simberloff/publication/216810728_The_Assembly_of_Species_Communities_Chance_or_Competition/links/553918270cf247b8588006a3/The-Assembly-of-Species-Communities-Chance-or-Competition.pdf) and (<https://besjournals.onlinelibrary.wiley.com/doi/10.1111/2041-210X.12502>). Therefore, we decided to keep our original term.

Comment 11: Figure 1b – were any of the pairwise comparisons between expert community and sound prediction significantly different (at any site)? If so, might be worth flagging which one(s). If not, perhaps saying so explicitly in the figure legend will help emphasize that the indices performed well.

Reply: Thank you for your valuable input. Using a linear mixed model, we have identified statistically significant differences, notably for pastures and old-growth areas. We have indicated these differences using asterisks to address your suggestion and provided relevant testing information per your recommendation.

Comment 12: L164-166 – I didn't follow the logic that overlap of composition between early and late-stage recovery suggests a variety of recovery trajectories. In fact, to me, it seems to suggest the opposite; that composition recovers rapidly and remains largely unchanged during later stages of recovery, unless I'm missing something? Perhaps worth recasting or expanding this point a little.

Reply: This is a good point. It shows that there is more change in the early phases of recovery and a slowing down in later stages. This makes sense, because the physical change in habitat is probably most dramatic in the early years. We revised the text accordingly. The revised text reads now (line 176-181): *“First, we found a substantial overlap of species composition between the early and late recovery stages, with a most pronounced change in the early phase. This demonstrates a rapid shift of species compositions after abandonment and a slowing down of community changes in the later stages.”*

Comment 13: L187 – I think “acoustic niches” is a more common term than “soundscape niches”. Consider rephrasing?

Reply: Thanks changed accordingly.

Comment 14: Figure 2b – is linear regression the best approach here? Just by eyeballing the data, it looks like a quadratic term might improve model fit. Was model selection and comparison used to fit these models or was there an a priori expectation of linearity? I could not really tell from the Methods section. Same goes for other linear model fits throughout.

Reply: We thank the reviewer for this comment, which helped to clarify our approach. We followed Ockham's razor, i.e. the scientific principle to use a minimum set of characteristics and relation types for explanation, as long as prior knowledge doesn't suggest more complex links (principle of parsimony). Our starting point of the analyses is the first axis of the species community composition, which reflects the levels of recovery in a linear way. Thus, the most parsimonious hypothesis for other factors with close relationship to community structure also assumes linearity. In addition, this simple approach avoids the risk of overfitting in a dataset of only 43 observations as well as the risk of being trapped by post-hoc hypotheses after intensive data inspection. We hope having clarified this point in the revised version of the methods section. It reads now in the main text:

Main Text (Line 147-148): As the main axis of community composition (Fig. 1a) revealed a linear gradient from pastures to old-growth forests, we used linear models in all further statistics as the most parsimonious approach.

Methods section (Line 478-482): In this way a major first and second axis could be calculated as usual in ordination. The score on the first axis was extracted as baseline for further modelling. As this axis revealed a linear gradient of community recovery (Fig. 1a), we generally hypothesized a similar relationship for other indicators, too, and applied linear models in subsequent analyses.

Comment 15: L208 – While an R-squared value of 0.39 is indeed something I'd consider to be generally “high explanatory power” in ecology (especially for species rich tropical systems), given the general readership of the journal, I wonder if it might be better to tone this down slightly? Those reading from an AI background may disagree that 0.39 is high explanatory power, for instance.

Reply: We fully agree and followed your suggestion toning down here; we changed from high to good for the insects.

Comment 16: L213-214 – I'd maybe clarify here that indices seem to be useful surrogates for non-vocalising fauna in *this* system, but I'm not convinced there's enough evidence (from other similar studies) to phrase this as a sweeping statement across ecosystems (as it currently reads).

Reply: We follow your advice and changed to “might be”, to avoid overstressing our findings from one tropical region only.

Comment 17: L216 – “Sound indices” is another competing term used to describe acoustic indices. Please change to acoustic indices (or soundscape indices) for consistency to avoid confusion. Also L264. It might be worth using find and replace for “index” and “indices” to catch all instances of this.

Reply: we changed according to your suggestion!

Comment 18: L222 – Significant predictors in what sense? I assume $p = 0.05$, but worth stating explicitly in the table legend.

Reply: Changed accordingly.

Comment 19: Table 1 – In the legend please briefly explain what the response variables are, and perhaps say something like “see Methods for explanations of acoustic index calculation”, in case for some reason a reader jumps straight to table 1.

Reply: Thanks, changed accordingly.

Comment 20: L227-229 – when citing figure 3 and table 1 here, it’s worth reminding readers how community vertebrate axis 1 relates to recovery time (i.e. habitat type). (same with Table 1 cited in L249).

Reply: We changed the first instance of this accordingly. The second (in line 249) was not focused on community vertebrate axis 1 but on richness.

Comment 21: L229 – Is fig S2 mis-cited here? Perhaps I’ve missed something.

Reply: Thanks for your feedback. Figure S2 was indeed mis-cited. Fig S3 is correct. We changed it accordingly.

Comment 22: Fig 3 – as with Fig 2, might be nice to include some measure of goodness-of-fit or similar, here. Also, the community vertebrate axis 1 is not that easy to interpret visually without referring to Fig 1. Might it be worth briefly explaining the loadings on this PC Axis in the legend or as simple labels at the Y axis value extremes?

Reply: We fully agree and added additional information to Fig 3 and to the Figure caption, as well as to caption of Fig. 2.

Comment 23: L243-246 – alternatively, might it be that mature forests have a greater richness or abundance of broadband insects which can saturate soundscapes (e.g. cicadas), rather than necessarily a broader diversity of species across taxa?

Reply: We agree that this could be a potential explanation; however, we could not find a corresponding theoretical reasoning for this, so for brevity, we did not include this additional alternative explanation.

Comment 24: L256 – repeated “that”, delete one.

Reply: We removed the second that.

Comment 25: L287 – I'm not sure I 100% agree that long-term acoustic data is easy to store. Often this data is very large and required a LOT of data storage. Maybe the point was that it's easy to store compared to type specimens? Or to Winkler/pitfall samples? Something else?

Reply: Thank you for your valuable input. We have modified the text to address the reviewer comments. It reads now (Line 311-314): *"The standardized collection of raw sound environmental data, such as soundscapes, creates a reproducible comprehensive long-term data basis in biodiversity monitoring that is easier to store in the long term than many specimen collections and largely independent of the collector."*

Comment 26: L288 – "open access strategy" is a bit too vague for me to understand exactly what is meant here. Also, as above, and in agreement with a previous reviewer, I don't quite see the link to greenwashing. Please remove or recast to better clarify.

Reply: We revised the text as follows to clarify our view of why this method will help to avoid greenwashing, It reads now (L314-318): *"This, in combination with making soundscape data publicly accessible, could also help to reduce greenwashing in carbon-focused conservation. Being able to directly quantify biodiversity, rather than relying on proxies such as growing trees, encourages and allows external assessment of conservation actions and promotes transparency."*

Comment 27: L292-293: Maybe rephrase to "the joint biodiversity and climate crises" or similar. Currently feels an odd formulation.

Reply: Thanks, changed accordingly.

Comment 28: End of comments. Congratulations on producing a manuscript that is important, interesting, and enjoyable to read.

Sam Ross

Reply: Thanks Sam for your detailed comments, which improved our manuscript. We added you to the Acknowledgement section.

Reviewer #5 (Remarks to the Author):

Comment 29: I have reviewed the manuscript titled "Soundscapes and artificial intelligence provide powerful tools to track biodiversity recovery in tropical forests" by Müller and colleague and I am providing some comments focusing on the insect metabarcoding parts, as per the editor's request. I think that the results presented in the MS are very promising and that it would be extremely useful to add soundscapes and artificial intelligence to the list of tools capable of tracking biodiversity recovery in tropical forests. Therefore, I think the work has the potential to be very significant in the field. I think that this work generally supports the conclusions and claims reported. Although I have some minor comments and questions that I hope may contribute to improve the overall MS.

Reply: Many thanks for the positive evaluation and the improvements to our manuscript

Comment 30: While I have not found any flaws in the data analysis, I think the interpretation and conclusions on the insect metabarcoding section can be made clearer. The authors appear to have a very interesting dataset, composed by almost 5000 Bins, which can provide a lot of information. I

was a bit disappointed as it seems the author did not really explore this dataset, as they did not provide almost any information on the insect diversity recorded. I think the author could address some of the points below and the MS could be improved substantially, with very little effort. I understand that insect metabarcoding was a relatively small component of this work, but considering the large volume of works published in the last few years, it would be important this work remains consistent with the standards of this field, by ensuring that the insect identification used here is robust. I have a couple questions for the authors, since I don't seem to find any answer in the manuscript.

Reply: We thank the reviewer for a careful consideration of our manuscript. We agree that the dataset from barcoding could be explored on its own; however, this is beyond the scope and word limit of this manuscript. Nevertheless, we provided a number of additional information in the text about the composition of the light trap data set at the level of orders. We further provide the raw fasta file public available now. Furthermore, we provide also the BIN-plot matrix used in the analyses including taxonomic information of the BINs and the direct link for each BIN to the respective bold homepage. This allows a quick check to all interested readers.

I was requested to revise the MS focusing on the topic of insect metabarcoding, however some aspects of the MS were not clear to me but important for my understanding of the sampling and analysis:

Comment 31: The authors have been sampling insects using a light trap, in particular a trap that has been produced mostly to attract Lepidoptera (LepiLED). The authors should probably state this can be considered a bias, since not all nocturnal insects are equally attracted to light. Perhaps the authors could not collect vocalising insects that were not attracted to light? This would have excluded some crucial taxa from their analysis.

Reply: We agree that light trapping has a clear bias towards moths and dipterans, however even other groups like song cicada and quite a wide range of other orders are collected with this method. So we can say this method has a bias (as all other methods as well), but on the other hand it is still one of the most effective methods to collect nocturnal insects in the tropics. For clarification we added a sentence to the methods as you recommended (line 423-425): “This kind of light trap attracts predominantly Lepidoptera and Dipteran. However, with the wide range of species attracted, it is one of the most efficient methods for tropical nocturnal insects, even collecting some vocalizing insect species as cicadas.”

Comment 32: The authors mention that only 13 species of insects were found to be vocalising nocturnal insects (line 210), over a total of almost 5,000 BINs. By removing these species, both indices and CNN still had a high explanatory power even for species that are not part of the vocalizing tropical animal community. Can the authors expand on this? How is it possible that soundscape indices and CNN work even if the insects are non-vocalising? I admit this being entirely outside of my area of expertise, but if the soundscape indices have strong explanatory power, couldn't it be because some of those >4000 BINs may actually include vocalising insects? Considering our scarce knowledge of the species present in high-biodiversity areas in Central and South America, how can the authors safely exclude all other BINs from being “vocalising insects”.

Reply: The ecological mechanism behind this pattern is that vocalizing vertebrates (birds, frogs, few mammals) similarly represent the gradient of recovery as the insect communities do. For most of our arthropod identification, we can guarantee that they do not vocalize at all or at frequencies as recorded by our sound recorders. As the majority of species are Lepidoptera and dipteran,

dominating the community composition pattern in our analyses, the small proportion of vocalizing insects is not able to change the results of our analyses.

Comment 33: Additionally, the authors should explain in the methods, which are the 13 vocalising insects they recorded and subsequently removed from the dataset for some of their analyses.

Reply: We agree and revised the text as following (Line 226-228): *“In further testing, we excluded the 13 vocalizing insect species from the families Tettigoniidae, Gryllidae, Cicadidae, present in our light traps, which did not change the axis of species composition (Pearson correlation $\rho=0.986$), indicating that light trap assemblages predominantly represent non-vocalizing insects.”*

In addition, I have provided a few corrections/comments I hope may improve the work:

Comment 34: Everywhere in MS. While it is true that there may be variant, the general consensus on the use of the word “metabarcoding” is to write it without hyphen. Please remove all instances of “meta-barcoding”.

Reply: We have changed this throughout the manuscript.

Comment 35: Line 130. The authors have used the BIN system as a proxy for species identification. While this system can be a very valid approach to assess and compare diversity, unfortunately it doesn't automatically translate to species identification. While the results are not reported in the main text, I doubt the authors have “identified all insects using metabarcoding”, since many (most?) of the taxa could probably not be identified to species level. This is normal with insect metabarcoding, especially in area with high level of endemism and diversity. I would suggest the author rephrase the sentence with something like: “...with autonomous light traps and assessed insect diversity using metabarcoding..”

Reply: We fully agree and changed the wording according to your suggestion. Please see also our reponse to comment 42/43 where we used the opportunity to clarify our BIN approach.

Comment 36: Line 256. Repetition of “that”. Remove the second “that”. “Our data show that not only nocturnal insects recover quickly.”

Reply: Thank you, we removed this repetition.

Comment 37: Line 390: Please, specify which model of LepiLED, since this can impact the number of LED lights on the lamp and therefore the different peaks in the light spectrum.

Reply: Thanks. In the revised manuscript, we now added more information on the LepiLED Mini Switch, UV-mode switched off.

Comment 38: Line 392. “Collections” should start with a lower case: “collections”.

Reply: Done.

Comment 39: Line 396. Please specify which alcohol. Was it ethanol?

Reply: We added the information that we used 96% undenaturated ethanol instead of alcohol.

Comment 40: Lines 397-400. The Skelton et al. 2022 reference is a very good work, however it is

focused on environmental DNA. Since you are working on insect metabarcoding from full specimens, Elbrecht & Leese as well as Martoni et al. 2022 are going more in depth on the issues linked to reads bias due to biomass, primers and taxonomic similarity. The authors should acknowledge that while size filtering is very good practice, this unfortunately cannot guarantee all insects have been recorded, and that reads bias has been avoided. Even closely related species belonging to the same genus may be differently impacted by the same DNA extraction/amplification/sequencing procedure, generating a difference in read numbers (i.e. a bias), that may result in loss of rare species; see Martoni et al. 2022.

Martoni F, Piper AM, Rodoni BC, Blackett MJ. 2022. Disentangling bias for non-destructive insect metabarcoding. PeerJ 10:e12981 <https://doi.org/10.7717/peerj.12981>

Reply: We fully agree and added this reference and more comments to the text. The optimization of the sequencing of insect bulks is still developing. In fact, the first author is involved in this progress with the team of Florian Leese. So far, there is no single best solution to balance the sequencing depth, thieving, splitting the samples and the financial effort in metabarcoding. Our approach from thieving to extraction to assigning sequences to BINs, is one currently very useful approach for ecological analyses, see also reply to comment 42. In the revised text, we add more nuance about the various methodological considerations. It reads now (Line 435-441): “Size filtering is only one tool to improve the balance between small and rare species on the one hand and large and abundant species on the other hand in bulk samples, and even this cannot guarantee all insects be detected. Here even species belonging to the same genus may be differently impacted by the same DNA extraction/amplification/sequencing procedure, generating a difference in read numbers, which may result in loss of rare species. However, our approach was standardized for all samples.”

Comment 41: Line 401. Full name of gene (cytochrome oxidase) should be italicised. Gene name should be reported as “Subunit I of the mitochondrial gene cytochrome oxidase”.

Reply: We changed this accordingly.

Comment 42: Lines 406-407. This is the issue I mentioned at line 130 linked to the BINs methodology. The fact that a BIN cluster matches a sequence at 90% means that species could NOT be identified. A 90% match of a short fragment of COI means the species is probably in the same genus, but certainly not the same species. As I mentioned before, this is not a methodological issue, but the authors should be clear about their results. I don't think it is fair to say that a match at 90% “allow comparison with studies based on morphological determination”. I would remove the sentence entirely after “genetic similarity)”

Reply: In response, we've addressed comments 42 and 43 together, seeing them as closely related from our perspective.

Comment 43: Lines 407-408. Why was the library limited to countries from Central and Southern America? And most importantly, how? This could be a relatively major issue. If the author limited the reference library to samples collected from Central and Southern America, they may have limited their own identification efficacy. If a species is present both in Argentina and in Australia, but was sequenced only from Australia, have the authors removed the data because the sequence is not from Central and Southern America? Therefore impairing their identification system by removing sequences that could in fact match 100% with their data? Or did the author only included species known to be present in Central and Southern America? If this

is the case, the issue could be minor. However, by doing so they won't be able to identify exotic species (i.e. not from Central and South America) that have been introduced to the area.

Reply: Thanks for the two important comments 42/43, which gave us the opportunity to clarify our approach and avoid any misunderstanding. We have modified the text to address the reviewer's main concerns. You are right that with our approach, a species not deposited with a sequence in Central or South America cannot be identified as such. However, preliminary analyses with global blasts showed that this can result in completely implausible species identifications. For example, using global blasts, species from Greenland appeared in the Choco Forest. Nevertheless, for the analyses in this manuscript, it is largely irrelevant whether we use global or regionally restricted approach, as they lead to only a slightly different "genetic morpho (BIN) species" assignment. To be more transparent now, we uploaded the sequence data and our BIN-plot matrix including the BOLD_HIT%ID for each BIN identified, as well as the direct link for each BIN to BOLD. The revised text reads now (Line 443-469): *The overall aim was to obtain a matrix of taxonomic units that closely resembles the concept of species. Therefore, the COI sequences were used to attribute Barcode Index Numbers (BINs), which are clusters of barcode sequences that can be used as a proxy taxonomic unit. BINs avoid the situation that in certain lineages there are unequally more OTUs even within a species. The allocation to BIN units is a challenge, given today's still very incomplete libraries. In many regions, corresponding libraries are largely missing for large species groups. This is true especially for groups harboring "dark taxa" such as dipterans, hymenopterans and hemipterans, but also for large portions of other arthropod species which are currently not referenced for South America. For ecological analyses, the goal is to assign the sequences to units representing the solution of species themselves and to derive ecological properties from the sequence information. For this purpose, we have developed the following procedure. The sequences are assigned to the next existing BIN from the studied and neighbouring countries reporting the genetic distance. This proximity and the information to which family, genus or species the sequence belongs is reported. Thus, BINs with a distance < are seen as identified species, while for those with a distance >3% function as "genetic morpho-species" in the ecological analyses (65). On this basis, all sequences across all lineages receive a reasonably balanced assignment to taxonomic units (and/or interim species identifications such as the BIN) and information on ecological properties, e.g. pollination (66), is provided at different levels (species, genus, family). By restricting the library to BINs with georeferenced records from Central and South America, we limit the references in such a way that, for example, a newly introduced species from New Zealand would not be recognised without a reference from South America. Conversely, blasts against the global database showed that a vast number of species identifications appear, which is completely implausible and complicates further ecological interpretations. The final data set of arthropods comprised 4557 BINs from 24 orders dominated by Lepidoptera (40%) followed by Diptera (32%), Hemiptera (10%), Hymenoptera (7%), Coleoptera (6%) and Trichoptera (2%).*

Comment 44: Line 408. The authors are reporting BINs from 39 orders. This suggests that they recorded more than just insects. What are the extra orders from? Arachnida? Mites? Would it be better to revise the manuscript and use "arthropods" instead of insects? Would it be possible to include some more information on the other orders? For example, "dominated by Lepidoptera (% of BINS) and Diptera (% of BINS), followed by Order (%), Order (%)."

Reply: Thanks for the comment. To clarify, we revised the text as follows (Line 135-2-139): *To test the generality of our results, i.e., that community composition indeed tracks faunal recovery, we iv) used a sound-independent dataset by sampling nocturnal insects with autonomous light traps and identified all arthropods by meta-barcoding. As fewer than 1% of species were non-Insecta, hereafter we refer to this dataset as insects.*

We appreciate your thorough assessment of our work. We've recognized that there was an error in the initial count of 39 orders, and in reality, we have 24 arthropod orders. We have taken immediate steps to rectify this mistake and will ensure that all necessary information is now provided (Line 466-469): *The final data set of arthropods comprised 4557 BINs from 24 orders dominated by Lepidoptera (40%) followed by Diptera (32 %), Hemiptera (10%), Hymenoptera (7%), Coleoptera (6%) and Trichoptera (2%).*

Comment 45: Line 411. Please reference R every time you use it in a manuscript:
R Core Team (2023). R: A language and environment for statistical computing. R Foundation for Statistical Computing, Vienna, Austria. <https://www.R-project.org/>.

Reply: We agree and added the reference in all sections when using R.

Comment 46: Line 515. Please, change “Next Generation Sequencing” with “High Throughput Sequencing”. During the past 5-7 years it has been pointed out how these technologies are not “next generation” anymore, since they have been around since 2007-2008, as the authors have used in the same paragraph.

Reply: We appreciate the suggestion. We changed the subtitle according to your suggestion.